# Salivary and plasmatic oxytocin are not reliable trait markers of the physiology of the oxytocin system in humans

Daniel Martins[1], Anthony S Gabay[1,2], Mitul Mehta[1,2], Yannis Paloyelis[1]*

[1]Department of Neuroimaging, Institute of Psychiatry, Psychology & Neuroscience, Kings College London, London, United Kingdom; [2]Centre for Human Brain Health, University of Birmingham, Birmingham, United Kingdom

**Abstract** Single measurements of salivary and plasmatic oxytocin are used as indicators of the physiology of the oxytocin system. However, questions remain about whether they are sufficiently stable to provide valid trait markers of the physiology of the oxytocin system, and whether salivary oxytocin can accurately index its plasmatic concentrations. Using radioimmunoassay, we measured baseline plasmatic and/or salivary oxytocin from two independent datasets. We also administered exogenous oxytocin intravenously and intranasally in a triple dummy, within-subject, placebo-controlled design and compared baseline levels and the effects of routes of administration. Our findings question the use of single measurements of baseline oxytocin concentrations in saliva and plasma as valid trait markers of the physiology of the oxytocin system in humans. Salivary oxytocin is a weak surrogate for plasmatic oxytocin. The increases in salivary oxytocin observed after intranasal oxytocin most likely reflect unabsorbed peptide and should not be used to predict treatment effects.

## Introduction

In the last two decades, a wave of studies have sought to investigate the role of oxytocin in human normal and impaired socio-affective behavior and cognition (*Donaldson and Young, 2008*; *Dyball and Paterson, 1983*). Several methodological approaches have been adopted to this effect. Predominantly these include: (i) the measurement of psychological or neurobiological outcomes after the intranasal administration of exogenous oxytocin compared to placebo (*Leng and Ludwig, 2016*); (ii) genetic association studies between variations in candidate genes of the oxytocin pathway and behavioral and brain phenotypes (*Onodera et al., 2015*; *Verhagen et al., 2020*); (iii) the assessment of associations between single measurements of the concentration of endogenous oxytocin in peripheral fluids (mainly blood and saliva) at rest (*Valstad et al., 2017*) and individual differences in neurobehavioral phenotypes (*Crockford et al., 2014*), psychiatric disorder status and/or symptom severity (*Rutigliano et al., 2016*).

The latter approach is based on two assumptions. The first is that single measures of baseline levels of endogenous oxytocin in the biological fluids of peripheral compartments can accurately index oxytocin release in the brain (*Valstad et al., 2017*). The second is that single measures provide reliable estimates of baseline levels of endogenous oxytocin in plasma or saliva. Definitive answers regarding the first assumption, whether it regards baseline or evoked release following some intervention, remain to be obtained (*MacLean et al., 2019*).

Here, we investigate the second assumption, which is a prerequisite if single measurements of baseline levels of endogenous oxytocin are to be used as a valid trait markers of the physiology of the human oxytocin system (*Wang et al., 2006*). Currently, we lack robust evidence that single measures of endogenous oxytocin in saliva and plasma at rest are stable enough to provide a valid

*For correspondence:
yannis.paloyelis@kcl.ac.uk

Competing interests: The authors declare that no competing interests exist.

trait marker of the activity of the oxytocin system in healthy individuals. Indeed, previous studies have claimed within-individual stability of baseline plasmatic and salivary concentrations of oxytocin in both adults and children based on moderate-to-strong correlations between salivary and plasmatic oxytocin concentrations measured repeatedly within the same individual over time using ELISA in unextracted samples (*Feldman et al., 2013*; *Schneiderman et al., 2012*; *Gordon et al., 2017*). However, these studies have a number of methodological limitations that raise questions about the validity of their main conclusion that baseline plasmatic and salivary concentrations are stable within individuals. First, measuring oxytocin in unextracted samples has been postulated as potentially erroneous, given the high risk of contamination with immunoreactive products other than oxytocin (*Szeto et al., 2011*). It is conceivable that these non-oxytocin immunoreactive products might constitute highly stable plasma housekeeping proteins (*Zhu et al., 2019*), which masked the true variability in oxytocin concentrations. Second, a simple correlation analysis cannot provide information about the absolute agreement of two sets of measurements – which would be a more appropriate approach to study within-subject reliability/stability. Third, it is not clear whether these findings generalize beyond the early parenting (*Feldman et al., 2013*) or early romantic (*Schneiderman et al., 2012*) periods participants were in when the studies were conducted, since these periods engage the activity of the oxytocin system in particular ways (*Gordon et al., 2010*). Hence, establishing the validity of salivary and plasmatic oxytocin as trait markers of the activity of the oxytocin system in humans remains as an unmet need. Such evidence is urgently required, given reports that plasma and saliva levels of oxytocin are frequently altered during neuropsychiatric illness and that they covary with clinical aspects of disease (*Rutigliano et al., 2016*).

The measurement of salivary oxytocin has been proposed as a surrogate for blood plasma levels (*Valstad et al., 2017*). Compared to blood sampling, saliva collection presents several logistical and measurement advantages (i.e. relatively clean matrix) (*Gröschl, 2008*). However, the validity of this approach remains unclear. While elevations in plasmatic oxytocin after intranasal administration are likely to result from capillary absorption in the nasal cavity (*Gossen et al., 2012*), elevations in saliva oxytocin could result from mucociliary clearance of intranasally delivered oxytocin from the nasal cavity to the oropharynx ('drip-down' oxytocin) (*Huffmeijer et al., 2012*; *van Ijzendoorn et al., 2012*). This question can be illuminated by ascertaining changes in salivary oxytocin following the intravenous administration of oxytocin, eliminating the confound of drip-down oxytocin. In a previous study using enzyme-linked immunosorbent assay, the administration of low and medium doses of intranasal oxytocin (8 and 24 IU) elevated oxytocin concentrations in saliva, but the administration of a low dose (1 IU) intravenously had no effect. Moreover, concentrations of oxytocin in plasma and saliva did not correlate at baseline or after the administration of exogenous oxytocin for either route (*Quintana et al., 2018*). These data suggest that salivary oxytocin is a weak surrogate measure for peripheral blood levels. However, questions remain about whether the same results would have been achieved by using a more sensitive method of quantification, such as radioimmunoassay (currently the gold-standard for oxytocin quantification [*McCullough et al., 2013*]), or higher doses of oxytocin administered intravenously.

Here, we aimed to characterize the reliability of both salivary and plasmatic single measures of basal oxytocin in two independent datasets, to gain insight about their stability in typical laboratory conditions and their validity as trait markers for the physiology of the oxytocin system in humans. Additionally, we investigated whether salivary oxytocin concentration reflects plasmatic oxytocin by examining (i) if the intravenous administration of a large dose of oxytocin which produces sustained increases in plasmatic oxytocin over the course of 2 hr also increases the concentration of salivary oxytocin; (ii) how potential changes in salivary oxytocin compare between different routes of administration (intranasal versus intravenous) and methods of intranasal administration (spray versus a nebuliser); and (iii) the correlation between plasmatic and salivary oxytocin levels at baseline and after the administration of exogenous oxytocin using two different methods of intranasal administration (spray versus nebuliser) and the intravenous route. For all analyses, we followed current gold-standard practices in the field and assayed oxytocin concentrations using radioimmunoassay in extracted samples, which has shown superior sensitivity and specificity when compared to other quantification methods (*McCullough et al., 2013*).

# Results

## Baseline salivary and plasmatic oxytocin concentrations across visits

We did not identify any significant differences in mean concentration of baseline oxytocin in saliva or plasma samples across the four visits of dataset A (Plasma: $F_{(1.47, 22.04)} = 0.51$, p=0.55; Saliva: $F_{(1.06, 12.82)} = 0.88$, p=0.38) (*Figure 1—figure supplement 1*). We also did not observe any differences in mean concentrations of baseline oxytocin in plasma across the two visits of dataset B ($T_{(19)} = 0.63$, p=0.54). However, in a quick inspection of *Figure 1*, we can observe that the levels of baseline salivary and plasmatic oxytocin fluctuated considerably from one visit to another in most individuals (*Figure 1*).

## Reliability of single oxytocin measurements in the plasma and saliva

### Dataset A

#### Between-visits reliability analysis across the four visits

We estimated the intra-class correlation coefficient (ICC) of single oxytocin measurements in saliva to be 0.23 and in plasma 0.29. The mean coefficient of variation (CV) was 63% for saliva and 57% for plasma measurements. We estimated the number of measurements that would have been required to achieve good reliability (ICC = 0.80) of a putative averaged measure to be 13.39 for saliva and 9.79 for plasma (*Table 1*).

#### Between-visits reliability analysis for each pair of visits

Detailed descriptions of the ICCs and CV estimated for each pair of the four visits included in our analysis of dataset A are presented in *Supplementary file 1* – Table 1. For plasma, we found higher estimates of ICC for the following pairs: visits 1–2: 0.80 and visits 3–4: 0.66 (*Supplementary file 1* – Table 1). The CVs were 31% for visits 1–2% and 45% for visits 3–4. The estimated ICC for any of the other pairs of visits was not significantly different from 0 (*Supplementary file 1* – Table 1). For saliva, we only found higher estimates of ICC for the pair visits 2–3: 0.82 (*Supplementary file 1* – Table 1). The estimated ICC for the remaining pairs was not significantly different from 0 (*Supplementary file*

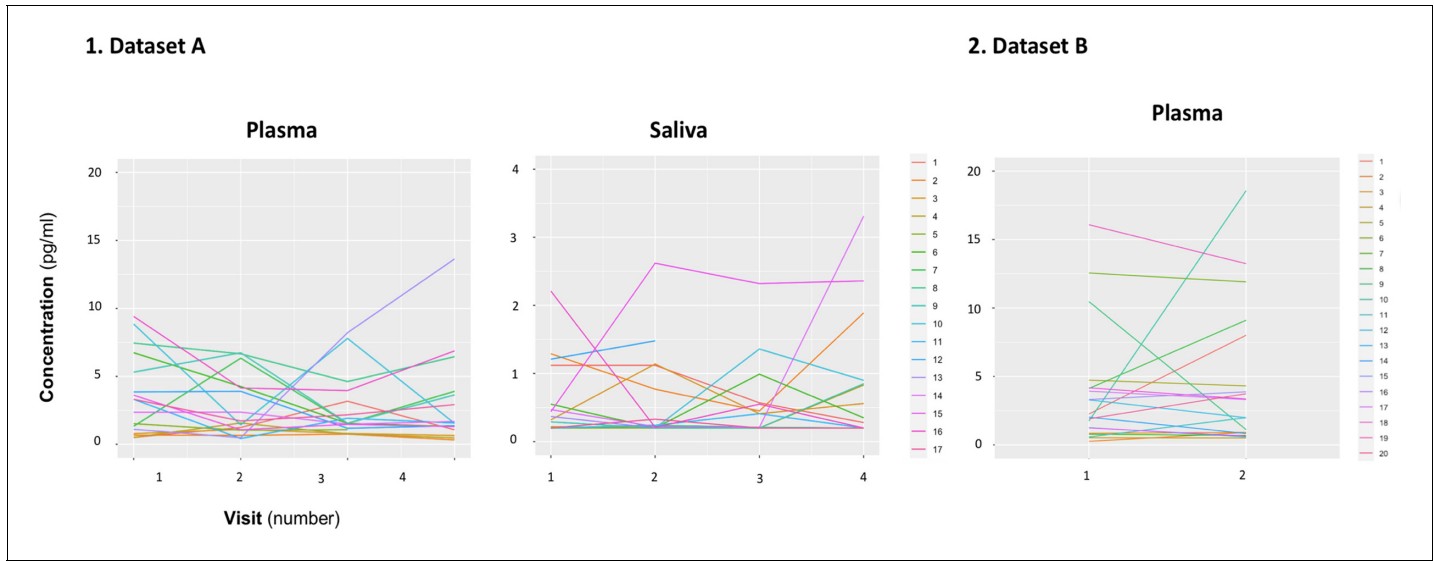

**Figure 1.** Within-individual variation of baseline measurements of oxytocin in plasma and saliva samples across visits. Baseline plasmatic and salivary oxytocin fluctuate from one visit to another for most individuals (Dataset A). We replicated this trend in an independent dataset for plasma (Dataset B). Each colored line represents one individual.

The online version of this article includes the following figure supplement(s) for figure 1:

**Figure supplement 1.** Mean plasmatic and salivary concentrations of oxytocin across visits.

**Figure supplement 2.** Correlation between baseline oxytocin concentrations in plasma (left) and saliva (right) for each pair of the four sessions included in Study A.

**Table 1.** Estimates of absolute and relative between-visits reliability of oxytocin measurements in saliva and plasma samples. Absolute and relative reliability were analyzed using the within-subject coefficient of variation (CV) and the intra-class correlation coefficient (ICC), respectively. We also present the number of additional measurements of the same individual that would be theoretically required to achieve different levels of reliability (ICC = X) of a hypothetical averaged measure, based on the initial reliabilities estimated for our datasets A and B. This number was calculated using the Spearman-Brown prediction formula. CI – confidence interval; SD – Standard Deviation; *H0: ICC is not significantly different from 0. N represents the actual size of the sample used to calculate the ICCs and the CVs.

| | | ICC | | | | | | Number of samples necessary to achieve an ICC = X of the averaged measurements in a design including multiple samples per individual | | |
| | | | 95% CI | | | | | | | |
| | | N | Lower | Upper | F test* | p-value | CV mean (SD) | X = 0.80 (good reliability) | X = 0.70 (moderate) | X = 0.50 (fair) |
| Study A | Plasma (N = 16) | 0.29 | 16 | 0.06 | 0.59 | F(15, 45) = 2.61 | 0.01 | 57 % | 9.79 | 5.71 | 2.45 |
| | Saliva (N = 13) | 0.23 | 13 | − 0.01 | 0.58 | F(12, 36) = 2.22 | 0.32 | 63 % | 13.39 | 7.81 | 3.35 |
| Study B | Plasma (N = 19) | 0.49 | 19 | 0.06 | 0.76 | F(18, 18) = 2.84 | 0.01 | 42 % | 4.16 | 2.43 | 1.05 |

1 – Table 1). Please see *Figure 1—figure supplement 2* and *Supplementary file 1* – Table 1 for correlations between baseline concentrations of oxytocin for each pair of visits of dataset A.

## Between-visits reliability analysis controlling for the time-interval between visits

In line with our main analysis, we found poor reliabilities for both salivary and plasmatic oxytocin in a subset of our sample where two consecutive saliva and plasma samples were collected with an exact gap of 7 days. For both plasma and saliva, the estimated ICCs were not significantly different from 0 and the CVs were 40% and 49%, respectively (*Supplementary file 1* – Table 2). Variance in the within-subject intervals between samples did not correlate with within-participant variance in oxytocin concentrations across participants neither for plasma (Spearman Rho = 0.406, p=0.118) or saliva (Spearman Rho = −0.524, p=0.065).

## Within-visit reliability (placebo visit)

We estimated the within-visit ICC to be excellent 0.92 and the CV to be 20% in the placebo session (*Supplementary file 1* – Table 3).

## Dataset B

We estimated the ICC to be 0.49 for single measurements of baseline plasmatic oxytocin across the two visits. The mean CV was 42%. The number of measures that would have been required to achieve good reliability of a putative averaged measure was estimated to be 4.16 (*Table 1*).

## Effects of intranasal and intravenous oxytocin administration on salivary and plasmatic oxytocin concentrations

For salivary oxytocin, we found a significant treatment × time interaction (F(3, 36) = 18.29, p<0.001). Post-hoc analyses revealed that the administration of oxytocin either by intranasal spray or nebuliser, but not the administration of intravenous oxytocin or placebo, resulted in significant increases of salivary oxytocin levels from baseline (Baseline vs Post-administration: Spray – t(12) = 7.06, adjusted p<0.001; Nebuliser - t(12) = 7.61, adjusted p<0.001; Intravenous - t(12) = 0.07, adjusted p=0.99; Placebo - t(12) = 0.15, adjusted p=0.99) (*Figure 2*).

For plasmatic oxytocin, we found a significant time × treatment interaction (F(3, 45) = 3.99, p=0.02). Post-hoc investigations revealed that the intravenous administration of oxytocin resulted in a significant increase in plasmatic oxytocin, but placebo or intranasal administration of oxytocin using either the spray or the nebuliser did not produce any changes from baseline at this time-point (Baseline vs Post-administration: Spray – t(15) = 1.38, adjusted p=0.52; Nebuliser - t(15) = 0.25,

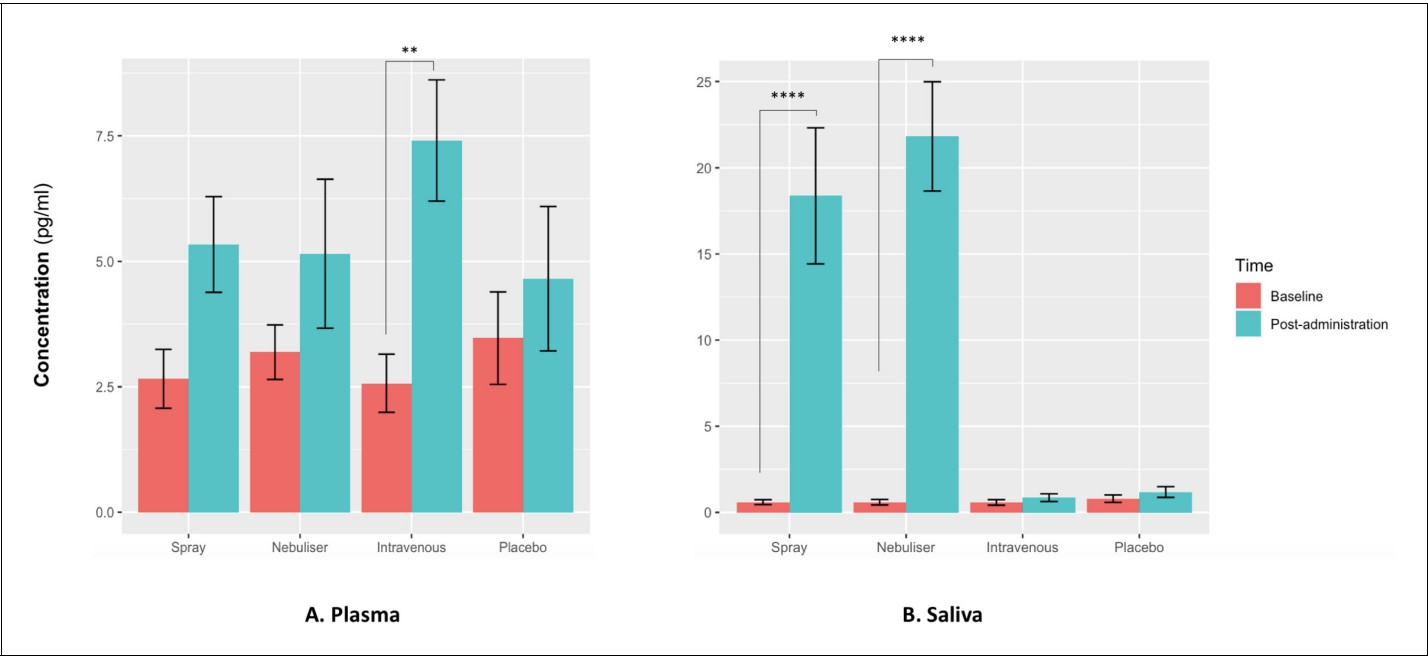

**Figure 2.** Effects of the administration of intranasal and intravenous oxytocin on salivary (A) and plasmatic (B) oxytocin. We examined the effects of treatment, time and treatment × time on salivary and plasmatic oxytocin in a two-way analysis of variance. Post-administration samples were collected at 115 min post-dosing. Statistical significance was set to p<0.05. **p=0.001 and ****p<0.001, using Tukey for multiple testing correction during post-hoc investigation of significant interaction effects. Please note that although all the statistical analyses were conducted on log-transformed oxytocin concentrations, here we plot the raw values to facilitate interpretation.

adjusted p=0.99; Intravenous - t(15) = 3.73, adjusted p=0.001; Placebo - t(15) = 1.54, adjusted p=0.41) (*Figure 2*).

## Association between salivary and plasmatic oxytocin at baseline and after administration of exogenous oxytocin

We did not find a significant correlation between oxytocin concentrations measured in saliva and plasma at baseline (r = 0.10 – Bootstrap 95% CI [−0.23,0.37], p=0.18, BF = 3.02, N = 63) (*Figure 3*) or following the administration of exogenous oxytocin (*spray*: r = −0.21 Bootstrap 95% CI [−0.64,0.32], p=0.43, BF = 2.43, N = 16; *nebuliser*: r = 0.07 Bootstrap 95% CI [−0.45, 0.57], p=0.65, BF = 3.32, N = 14; *intravenous*: r = −0.05 Bootstrap 95% CI [−0.52, 0.43], p=0.84, BF = 3.27, N = 17; *placebo*: r = −0.20 Bootstrap 95% CI [−0.64, 0.32], p=0.45, BF = 2.48, N = 16) (*Figure 4*). Changes in oxytocin concentrations from baseline to post-administration also did not correlate between saliva and plasma, in any of our treatment conditions (spray: r = 0.134 – Bootstrap 95% CI [−0.25,0.67], p=0.62, BF = 4.68, N = 16; *nebuliser*: r = 0.10 Bootstrap 95% CI [−0.50, 0.45], p=0.73, BF = 4.71, N = 14; *intravenous*: r = 0.08 Bootstrap 95% CI [−0.36, 0.68], p=0.76, BF = 5.19, N = 17; *placebo*: r = −0.16 Bootstrap 95% CI [−0.39, 0.27], p=0.56, BF = 4.47, N = 16).

## Discussion

Using radioimmunoassay, a gold-standard method for oxytocin quantification, we showed that i) single measures of baseline oxytocin concentrations in saliva and plasma are not stable within the same individual across different days; (ii) for plasma, we replicated this finding in an independent cohort; (iii) intranasal administration of exogenous oxytocin, despite method of administration, increases salivary oxytocin, but intravenous administration of a considerable dose does not produce any changes; iv) salivary and plasmatic oxytocin do not correlate with each other either at baseline or after the intranasal or intravenous administration of exogenous oxytocin. We discuss our main findings below.

We reported poor reliability indexes for single measurements of baseline oxytocin in both plasma and saliva. This suggests that single measures of baseline oxytocin in either fluid cannot be

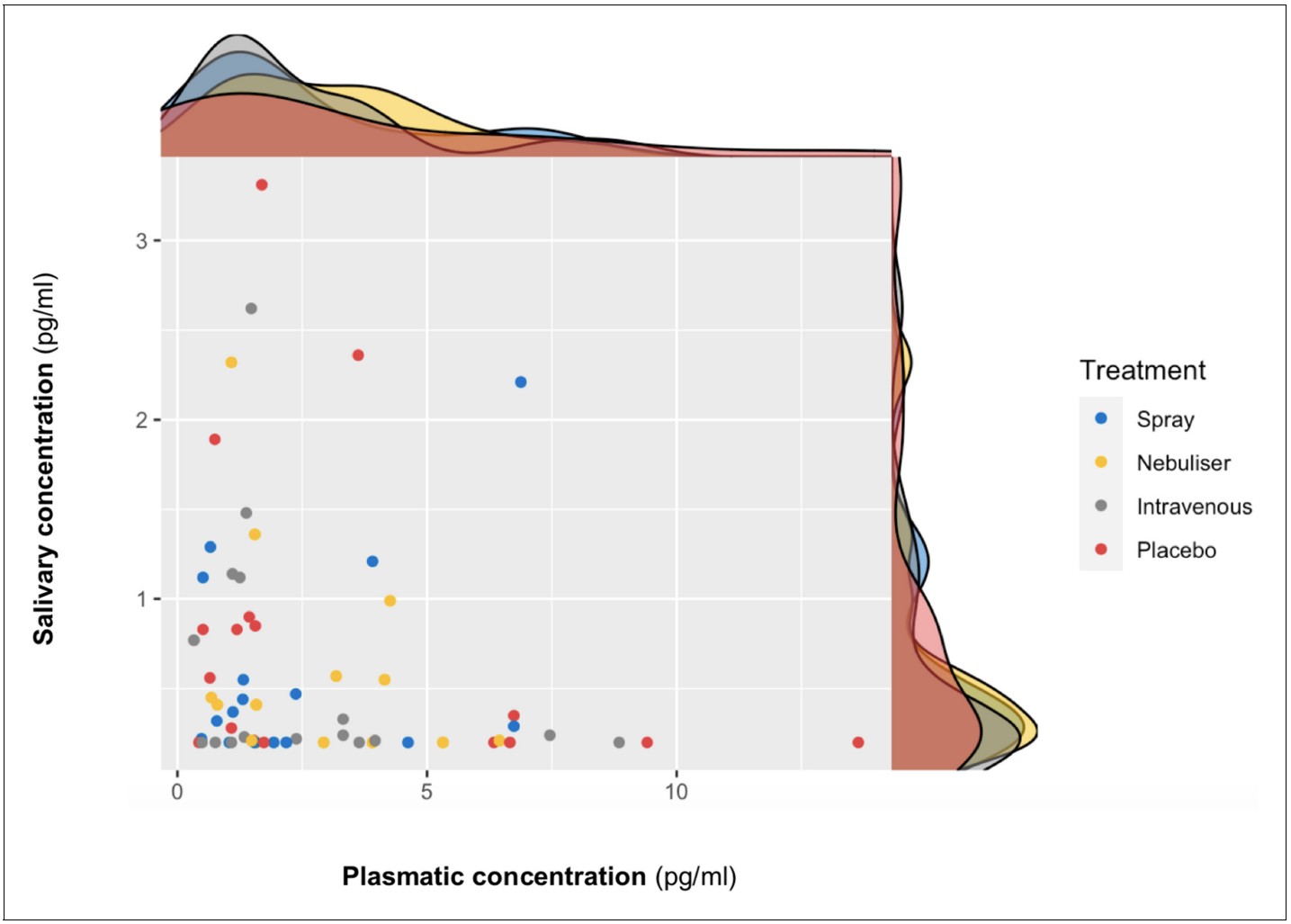

**Figure 3.** Association between salivary and plasmatic oxytocin concentrations at baseline. In this scatter plot, we depict the lack of association between salivary and plasmatic oxytocin concentrations at baseline (before any treatment administration). The density plots on the top of each axis show the distribution of oxytocin concentrations for each treatment level. Please note that although all the statistical analyses were conducted on log-transformed oxytocin concentrations, here we plot the raw values to facilitate interpretation.

consistently measured for the same individual across different days. The reliability estimates found in the current study are in the range of those already described for other hypothalamic hormones such as vasopressin (*Quintana et al., 2017*) or prolactin (*Muti et al., 1996*). These poor reliabilities are unlikely to be explained by variability in the time-interval between visits of the same individual. Three lines of converging evidence support this conclusion. First, we also found poor reliability indexes for both saliva and plasma when we restricted our analysis to a subset of our sample controlling for the exact number of days spacing visits. Second, we did not find any significant effect of time-interval on our estimated ICCs. If time-interval was driving the poor reliabilities then we would have expected that in our pairwise analyses reliability would be consistently higher for samples closer in time and drop as the time-interval between sessions increases. This was not what we found. Third, variability in the within-subject intervals between samples did not correlate with within-participant variance in oxytocin concentrations across participants.

We note that the mean CV for baseline concentrations of oxytocin in saliva and plasma is higher than four times the intra-assay variability of the radioimmunoassay we used (<10%). Furthermore, in a further analysis assessing the within-session stability of plasmatic oxytocin using two measurements collected 15 min apart from each other in the placebo visit (one sample collected at baseline and

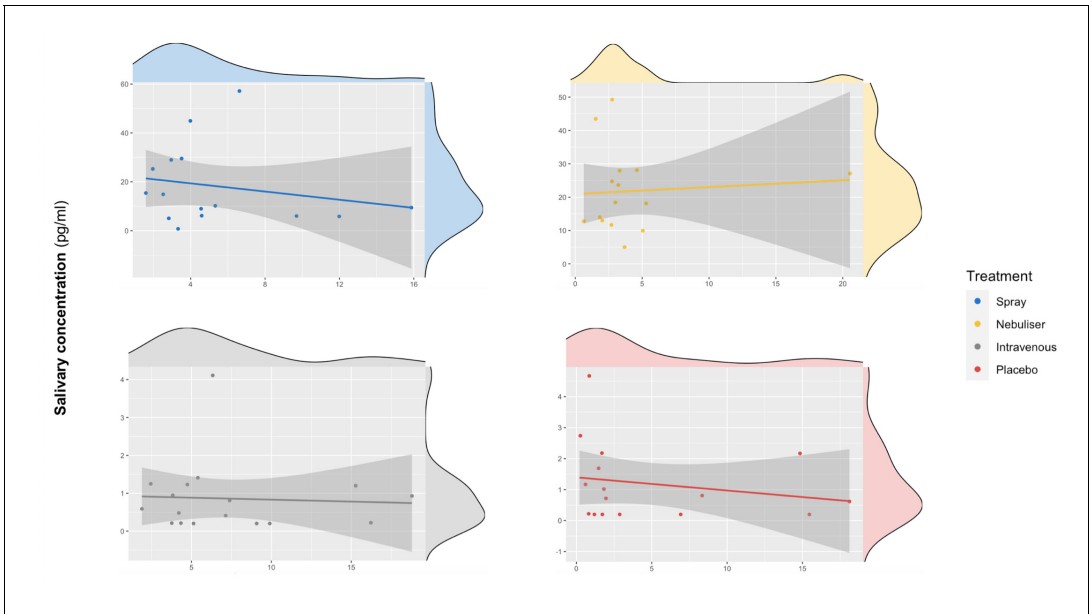

**Figure 4.** Association between salivary and plasmatic oxytocin after administration of intranasal and intravenous exogenous oxytocin. In these scatter plots, we depict the lack of association between salivary and plasmatic oxytocin concentrations at 120 and 115 min after the administration of intravenous and intranasal oxytocin or placebo, respectively. Each panel depicts data from one out of the four treatment levels. Please note that although all the statistical analyses were conducted on log-transformed oxytocin concentrations, here we plot the raw values to facilitate interpretation.

the other after the intravenous administration of saline), we found excellent within-session reliability (ICC = 0.92, CV = 20%). Together, this suggests that the low reliability of endogenous oxytocin measurements across visits in the current study results from true intrinsic individual biological variability and not technical variability/error in the method used for oxytocin quantification.

Although we made efforts to minimize variability in the conditions in which biological samples were collected across sessions in both datasets (i.e. asking participants to abstain from exercise the day before study; controlling the time of sample collection; processing samples immediately after collection to minimize differences in peptide degradation between samples), several additional factors might underlie the intra-individual biological variability in plasmatic and salivary oxytocin that we report here. These include differences across participants in sexual activity in the days preceding each visit, perceived stress, the amount of social interactions, or other non-acknowledged biological rhythms conditioning variations in oxytocin secretion throughout time (*Jong et al., 2015*; *Engel et al., 2019*). For instance, one study showed that oxytocin is secreted in a pulsatile manner in healthy males even at rest (*Baskaran et al., 2017*). Therefore, sampling during different phases of the pulsatile release of oxytocin for plasma across sessions could explain the discrepancies in oxytocin measurements observed in the current study across visits. The time-interval between measurements do not seem to significantly impact on the reliability of baseline oxytocin, as suggested by the overlap of the 95% confidence intervals of the ICCs estimated for each pair of sessions. Although higher reliabilities could be identified between some consecutive pairs of sessions (separated by an average of about 6 days from each other: sessions 1–2 and sessions 3–4 for plasma measurements), other consecutive pairs of sessions (for instance, sessions 2–3) did not produce significant reliability estimates. If there are specific reasons explaining the higher reliability indices observed for the specific pairs of sessions, these reasons remain to be elucidated. However, it is not implausible that we might have found higher reliabilities for these specific two pairs by chance, since the 95% confidence intervals for the ICCs for all pairs of samples overlapped.

Our observation of poor reliability questions the use of single measurements of baseline oxytocin concentrations in saliva and plasma as valid trait markers of the physiology of the oxytocin system in humans. Instead, we suggest that, at best, these measurements can provide reliable state markers within short time-intervals (5 min in our study). Our data does not support previous claims of high stability of plasmatic and salivary oxytocin within individuals over time. For instance, in one study,

*Feldman et al., 2013* assessed plasmatic oxytocin in recent mothers and fathers at two time-points spaced 6 months apart during the postpartum period. The authors found strong correlations between the two assessments for both mothers and fathers (*Feldman et al., 2013*). In another study, *Schneiderman et al., 2012* found strong correlations between plasmatic oxytocin concentrations measured at two different instances spaced 6 months apart in both single and individuals recently involved in a new romantic relationship (*Schneiderman et al., 2012*). Two important differences between these studies and ours are i) the method used for oxytocin quantification and (ii) the particular states participants were in when the studies were conducted. Regarding the first difference, these previous studies used ELISA without extraction, reporting concentrations of plasmatic oxytocin well above the typical physiological range of 1–10 pg/ml detected in extracted samples (in their studies, the authors report concentrations above 200 pg/ml). The inclusion of extraction has been postulated as a critical step for obtaining valid measures of oxytocin in biological fluids (*Szeto et al., 2011*). Unextracted samples were shown to contain immunoreactive products other than oxytocin (*Szeto et al., 2011*), which contribute largely to the concentrations of oxytocin estimated by this method. It is possible that these non-oxytocin products might represent highly stable plasma house-keeping molecules (*Zhu et al., 2019*) that masked the true biological variability in oxytocin concentrations between assessments in these previous studies that we could detect in extracted samples in our study. Regarding the second difference, these previous studies on within-individual stability were conducted during the early parenting (*Feldman et al., 2013*) or early romantic (*Schneiderman et al., 2012*) periods, which engage the activity of the oxytocin system in particular ways (*Gordon et al., 2010*). Instead, we used a normative sample that did not specify these inclusion criteria. Hence, we cannot exclude that during these specific periods the reliability of salivary and plasmatic oxytocin concentrations might be higher. We note though that our sample more closely resembles the samples used the vast majority of studies in the field (which sometimes even exclude participants during early parenthood [*Bui et al., 2019*). Hence, our estimates of reliability are a better starter point for all studies where specific circumstances potentially affecting the activity of the oxytocin system have not been specified a priori.

Our data poses questions about the interpretation of previous evidence seeking to associate single measurements of baseline oxytocin in saliva and plasma with individual differences in a range of neuro-behavioral or clinical traits. Almost all these previous studies have relied on the collection of single samples from each individual. Our findings suggest measures of oxytocin could be inconsistent and thus it is unlikely that a single sample may accurately represent oxytocin physiology and thus capture relevant inter-individual differences. Reliability of measurements also impacts power to detect correlations against these measurements (*Kanyongo et al., 2007*). To illustrate this point, in *Figure 5* we provide the results of a set of simulations investigating how different reliabilities of oxytocin measurements impact the sample sizes necessary to detect significant correlations between oxytocin concentrations in peripheral fluids and neurobehavioral outcomes of different effect sizes. Given the less-than-perfect reliability of oxytocin measurements (ICC around 0.30) we show here, to detect the small-to-medium (r = 0.20–0.50) correlations typically reported in studies using these measurements (*Torres et al., 2018*), researchers would need sample sizes between 102 and 651 participants to reach minimally acceptable power (80%). Sample sizes in association studies of endogenous oxytocin measurements are typically below 100 participants (*Torres et al., 2018*).

Turning to our second main finding, salivary and plasmatic oxytocin did not correlate at baseline or after administration of exogenous oxytocin (irrespective of route), replicating previous observations of a null association between measurements in these two compartments (*Javor et al., 2014*). Furthermore, we could not find any significant correlation between changes in salivary or plasmatic oxytocin from baseline to 115 min after the end of our last treatment administration in any of our four treatment conditions. The lack of significant associations between salivary and plasmatic oxytocin (and respective changes from baseline) was further supported through our Bayesian analyses which demonstrated that given our data the null hypotheses were at least three times more likely than the alternative hypothesis. Two hypotheses could account for the lack of correlation between plasmatic and salivary oxytocin. First, as in all other studies in the field, we did not control or manipulate the rate of saliva flow in the current study. Lipid insoluble molecules, such as oxytocin, enter into saliva mainly via the tight junctions between acinar cells through ultrafiltration (*Vining et al., 1983*). If oxytocin does reach the saliva compartment through an ultrafiltration mechanism (which depends on saliva flow rate [*Gröschl, 2008*]) then it is possible that when saliva flow is stimulated,

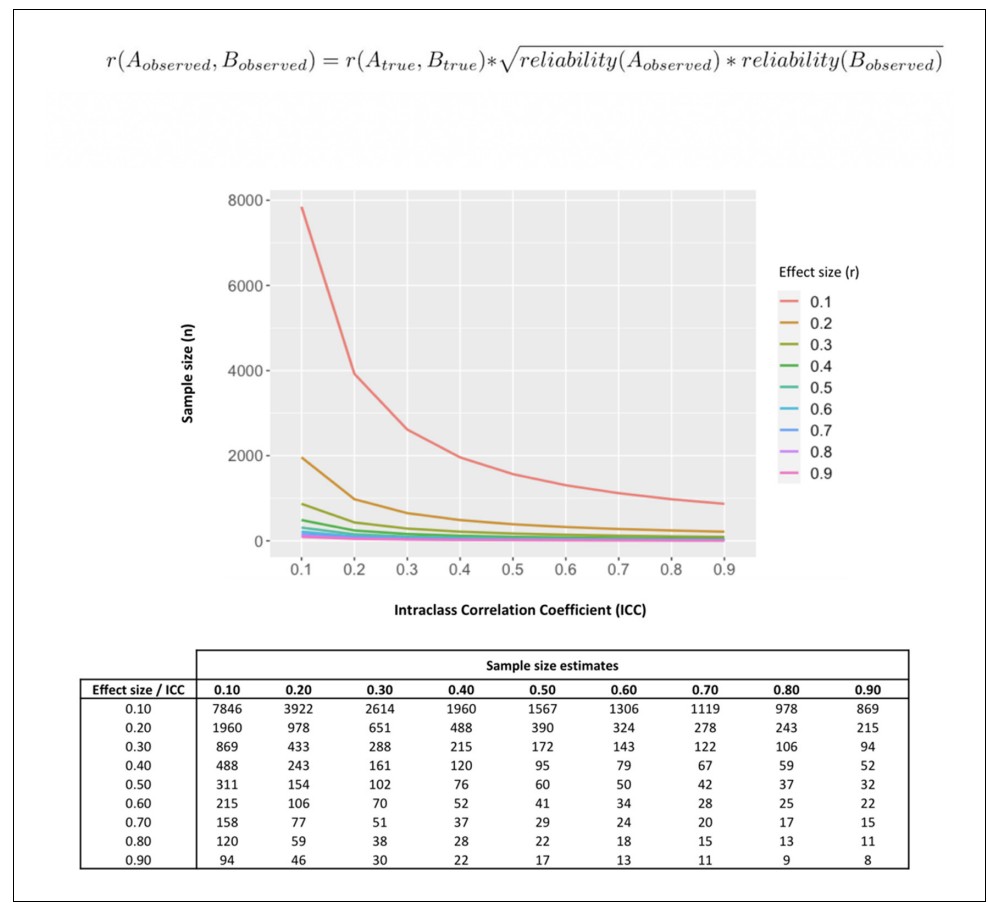

**Figure 5.** Influence of less-than-perfect reliability of oxytocin measurements on the sample sizes required to detect significant endogenous oxytocin-outcome associations in neurobehavioral human oxytocin research. In this figure, we show that the results of a set of simulations illustrating the impact less-than-perfect reliabilities of endogenous oxytocin measurements in peripheral fluids might have on the sample size required to detect significant oxytocin (**A**) – outcome (**B**) correlations of varying effect sizes in research studies. We conduct calculations for a minimally acceptable statistical power of 80% in a two-tailed parametric test. The outcome measure (**B**) was assumed to present perfect reliability. The figure was generated using the 'pwr.r.test' function of the 'pwr' R package. We specified 'r' according to the attenuation formula below (**Nunnally, 1970**). ICC -. Intraclass correlation coefficient; r – Pearson's correlation coefficient.

oxytocin measurements may better index its plasmatic concentrations. Second, we cannot discard differences in oxytocin degradation rates between saliva and plasma. Future studies should investigate these hypotheses further.

Studies have been using increases in salivary oxytocin after the intranasal administration of exogenous oxytocin to index systemic absorption and establish putative time-windows during which neurobehavioral effects of oxytocin administration may be expected. If oxytocin increases after intranasal administration reflected systemic absorption and transport from the blood then we would have expected that intravenous oxytocin would have also increased salivary oxytocin. Moreover, we would also have expected that increases in plasmatic and salivary oxytocin after intranasal administration would reflect the typical ratio of their concentration as observed during baseline (with lower concentrations in saliva than in plasma). Our findings were not consistent with these expectations. We could replicate previous evidence that intravenous oxytocin does not increase salivary oxytocin (**Quintana et al., 2018**) and extended it by showing that the lack of increase in salivary oxytocin is not limited to the specific low dose of intravenous OT that was previously used (1IU) and that it is not driven by the insufficient sensitivity of the OT measurement method which had resulted in more than 50% of the saliva samples being discarded in the previous study (**Quintana et al., 2018**). Therefore, our data supports the notion that increases in salivary oxytocin after its intranasal

administration most likely reflect drip-down oxytocin from the nasal cavity (*Quintana et al., 2018*). If this is the case then oxytocin elevations in saliva are mainly driven by non-absorbed exogenous oxytocin and therefore their use to estimate levels of systemic absorption or predict treatment effects after intranasal oxytocin administration is not valid. We expect this phenomenon to be particularly pronounced for higher administered volumes. Further studies should examine the impact of different administered volumes on increases in salivary oxytocin.

The lack of increase in salivary oxytocin after the intravenous administration of exogenous oxytocin that was consistently found in our study and in a previous study (*Quintana et al., 2018*) also raises the question of how oxytocin reaches the saliva if not from the blood. Currently, there is no evidence of direct acinar secretion or direct nerve terminals release of oxytocin to the saliva; therefore, transport from the blood remains as the most plausible mechanism of appearance of oxytocin in the saliva. Clarifying these mechanisms of transport is paramount, given the current hypothesis that salivary oxytocin might be superior to plasma in indexing central levels of oxytocin in the CSF (*Martin et al., 2018*).

One may argue the absence of significant elevations of oxytocin in saliva after its intravenous administration may be explained by significant differences in the kinetics of oxytocin concentration variation between these two compartments. This may include a significant delay in the elevation of oxytocin in saliva after the beginning of its increase in the plasma (explaining why saliva and plasma concentrations of oxytocin are not correlated after its exogenous administration). While this may be possible, in a previous companion paper we showed that the peak in plasmatic oxytocin occurs immediately after the end of its intravenous administration – 115 min before our post-administration saliva sample collection (*Martins et al., 2020*). It is therefore unlikely that this large interval of time would not have captured a significant delay in saliva elevation of oxytocin if it really existed, especially when after this time-interval plasma oxytocin still remained elevated, compared to baseline levels.

A strength of our study is the replication of poor reliability for baseline plasmatic oxytocin in a second independent dataset, which strengths our confidence in the robustness of our reliability findings. However, we acknowledge the following limitations. First, we only considered baseline measures in our reliability analyses. As for other hypothalamic-pituitary-adrenocortical markers where evoked measures present higher reliability than baseline measures (*Coste et al., 1994*), stimuli-evoked release of endogenous oxytocin (i.e. after social interaction, stress, pain) might also present higher reliability. Our conclusions are also restricted to male participants and to the radioimmunoassay method of oxytocin measurement, precluding extrapolations to female populations or to other quantification methods. Also, due to time and logistical constrains during our MRI setup, we could only sample saliva before administration and at the end of the scanning period, leaving our analyses on the effects of the administration of exogenous oxytocin on its concentration in saliva restricted to one single-time point post-administration. It is possible that we may have missed peak increases in saliva oxytocin after the intravenous administration of exogenous oxytocin if they occurred between treatment administration and post-administration sampling. This is unlikely given that the dose we administered intravenously resulted in sustained increases in plasmatic oxytocin over the course of 2 hr. Unless the half-life of oxytocin in saliva is much shorter than in the plasma, it would be surprising to not find any increases in salivary oxytocin after intravenous oxytocin given that concentrations of oxytocin in the plasma were still elevated at the specific time-point of our second saliva sample. Currently, we have no estimate for the half-life of oxytocin in saliva; however, given that previous studies have found evidence of increased salivary oxytocin after single intranasal administrations of 16IU and 24IU oxytocin up to seven hours post-administration (*van Ijzendoorn et al., 2012*), it is unlikely that the half-life of oxytocin is shorter in the saliva than in the plasma.

In summary, single measurements of baseline levels of endogenous oxytocin in saliva and plasma are not stable in typical laboratory conditions and therefore their validity as trait markers of the physiology of the oxytocin system is questionable. Salivary oxytocin is a weak surrogate for plasmatic oxytocin; hence, salivary and plasmatic oxytocin should not be used interchangeably. Finally, increases in salivary oxytocin after the intranasal administration of exogenous oxytocin most likely represent drip-down transport from the nasal to the oral cavity and not systemic absorption. Therefore, increases in salivary oxytocin after intranasal oxytocin administration should not be used to predict treatment effects.

## Materials and methods

### Participants

Dataset A included 17 healthy, right-handed, male volunteers (mean age (SD) = 23.75 (5.10); range = 18–34) who contributed samples over four visits, as part of a larger study. Dataset B (independent replication study) included 20 healthy, right-handed, male volunteers (mean age (SD) = 24.8 (3.70); range = 21–37), who contributed samples over two visits as part of a different study (both studies described below). All participants had no history of psychiatric disorders or substance abuse, scored negatively on a screening test for recreational drug use, and did not currently use any medication. In dataset A, we screened participants for psychiatric conditions using the Symptom Checklist-90-Revised (*Ruis et al., 2014*) and the Beck Depression Inventory-II (*Sacco et al., 2016*) questionnaires. In dataset B, we used the MINI International Neuropsychiatric Interview (*Sheehan et al., 1998*). Participants were advised to avoid heavy exercise, alcohol, or smoking the day before scanning and avoid any drink or food within 2 hr before scanning. Both studies were approved by King's College London Research Ethics Committee (Dataset A: PNM/13/14–163; Dataset B: PNM/14/15–32). For both studies, our sample size and number of samples collected per individual would have allowed us to detect intra-class correlation coefficients (ICC) of at least 0.70 (moderate reliability) with 80% of power (*Bujang and Baharum, 2017*).

### Study design

#### Dataset A

Participants were recruited to participate in a double-blind, placebo-control, triple-dummy, crossover MRI study exploring the effects of various methods of exogenous oxytocin administration on cerebral physiological responses at rest (*Martins et al., 2020*). Participants received, in counterbalanced order, over four consecutive visits spaced on average 8.80 days apart (SD 5.72; range 3–28), approximately 40IU of intranasal oxytocin, either with a nasal spray or the SINUS nebulizer (PARI GmbH), 10IU of oxytocin intravenously, or placebo. The administration of 10IU of oxytocin intravenously produces sustained increases in the levels of plasmatic oxytocin over a 2 hr course (*Martins et al., 2020*). This aspect of our design would allow us to discard the possibility that lack of changes in salivary oxytocin are due to under dosing. In each visit, blood samples were collected at baseline, immediately after each treatment administration, and at six time points post-administration, with the last sample acquired when participants came out of the scanner at about 115 min post-administration (*Figure 6*). Saliva samples were acquired at baseline and together with the last blood sample. For the purposes of this report, we use the plasmatic and salivary oxytocin measurements that were obtained at baseline and at 115 min after the end of our last treatment administration (this means that our post-administration samples were collected 115 min after the intranasal administrations and 120 mins after the intravenous administration of oxytocin). The full time course of changes in plasmatic oxytocin after the administration of intranasal and intravenous oxytocin in this study has been reported elsewhere (*Martins et al., 2020*).

All visits were conducted during the morning to avoid the potential confounding of circadian variations in oxytocin levels (*Amico et al., 1989*; *Reppert et al., 1984*). In addition, we also made sure that each participant was tested at approximately the same time across all four visits (all participants were tested in sessions with less than one hour difference in their onset time, except for one participant where the difference in the onset of one session compared to the other three sessions was 1.5 hr). All visits were identical in structure (duration ~3.5 hr). Upon arrival, participants gave consent and completed the required questionnaires. We then fitted an intravenous cannula on each arm of our participants (one for the intravenous administration and another for blood-sampling). Treatment was administered according to the study protocol (*Figure 6*). At the end of the treatment administration, participants were taken to an MRI scanner where we obtained a number of resting state and structural scans over the course of the next 2 hr.

#### Dataset B

Participants were recruited to participate in a study examining the effects of MDMA on social cognition (*Gabay et al., 2019*). Briefly, baseline blood samples were obtained 15 min before MDMA/

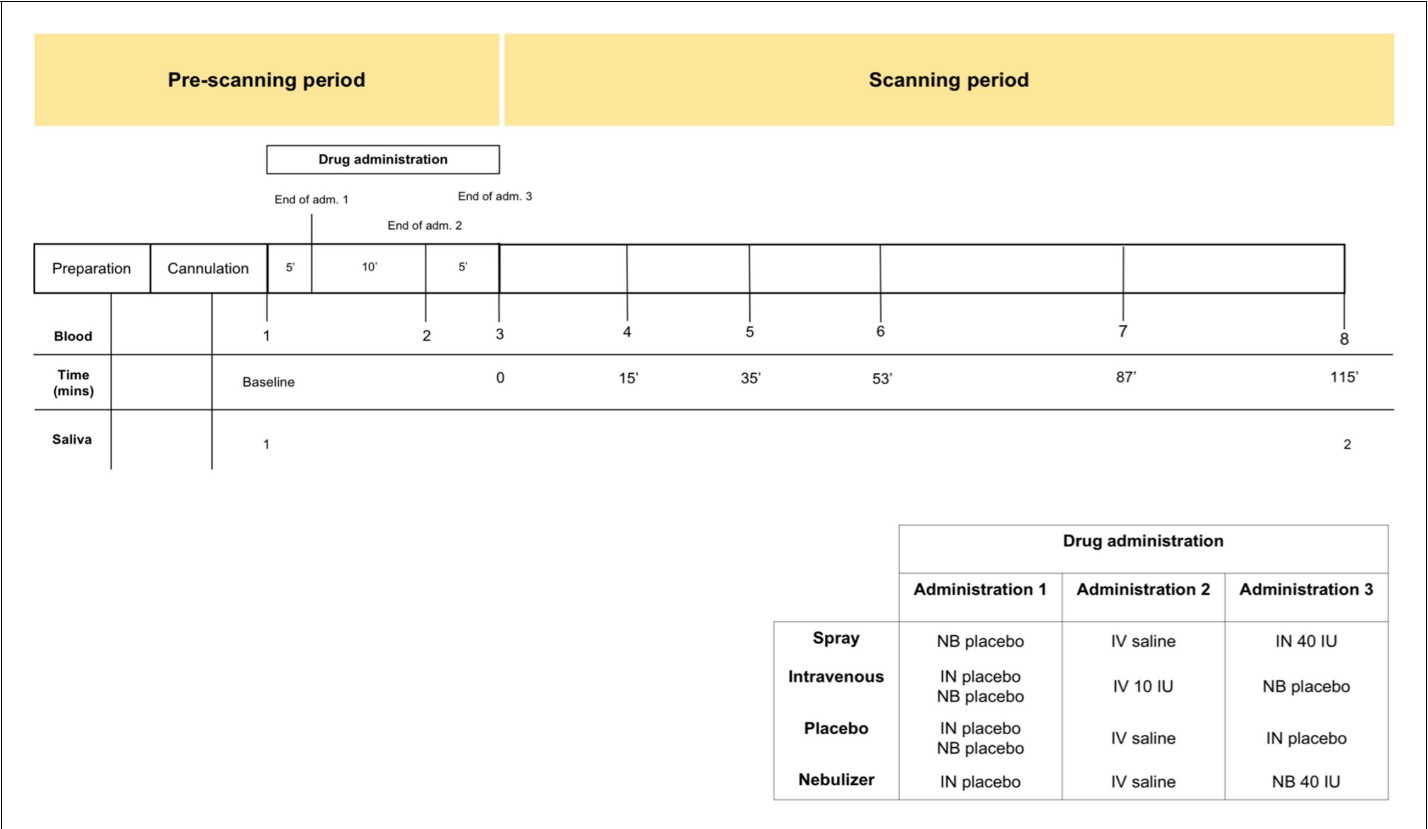

**Figure 6.** Schematic representation of the design of study A. All subjects received first an administration of intranasal placebo - either by spray or nebuliser, then an intravenous administration of oxytocin (10 IU)/saline and then an intranasal administration of oxytocin (40 IU)/placebo, either by spray or nebuliser. Following drug administration, participants were placed in a Magnetic Resonance Imaging scanner for eight resting arterial spinal labeling (ASL) regional blood flow images of the brain and one resting BOLD fMRI scan. Saliva samples were collected before any drug administration (baseline) and after the scanning session (at 115 min after our last treatment administration). Plasma samples were collected before any drug administration, after any administered drug and then at several time-points during scanning session. For the current study, only time-points where saliva and plasma were concomitantly collected were considered – baseline and after scanning session. Detailed plasmatic pharmacokinetics of each route/method have been presented elsewhere (*Martins et al., 2020*). Adm. – administration; min – minutes; IN – Spray; NB – Nebuliser; IV – Intravenous.

placebo administration on two separate occasions, spaced on average 9.30 days apart (SD = 5.70 days; range: 7–31 days).

## Saliva and plasma collections

Blood was collected in 5 ml ethylenediaminetetraacetic acid vacutainers (Kabe EDTA tubes 078001), placed in iced water and centrifuged at 1300 × g for 10 min at 4°C within 20 min of collection and then 0.5 ml of plasma was immediately pipetted into 2 ml Eppendorf vials. Samples were immediately stored −80°C until analysis. Saliva samples were collected using a salivette (Sarstedt 51.1534.500). Participants were instructed to place the swab from the Salivette kit in their mouth and chew it gently for 1 min to soak as much saliva as possible. After this, the swab was then returned back to the Salivette, centrifuged, 0.5 ml of saliva was aliquoted to 1.5 ml Eppendorf vials and then stored in the same manner as blood samples. *Salivettes* allow for a collection of mean saliva volumes in the range of 1.1 ± 0.3 ml (according to the manufacturer); high recovery of the concentrations of small peptides in saliva are consistently achieved when the sampled volumes are larger than 0.25 ml (*Gröschl et al., 2008*; *Harmon et al., 2007*). For both saliva and plasma, we followed the RIAgnosis standard operating procedures.

We followed this strict protocol, putting all samples in iced water until centrifugation with immediate storage at −80°C until analysis to minimize the impact putative differences in degradation of the peptide related to differences in the processing of the samples might have on the reliability of

the estimated concentrations of oxytocin. Minimizing the time-interval samples were kept in the collection devices also allowed us to keep potential absorption to the walls of these recipients to a minimum (*Gröschl et al., 2008*).

## Quantification of oxytocin in plasma and saliva samples

For both datasets, plasma and saliva oxytocin levels were analysed by a third party (RIAgnosis, Munich, Germany) using a Radioimmunoassay (RIA), as previously described (*Kagerbauer et al., 2013*). Plasma samples were extracted before quantification. Saliva samples were not extracted prior to quantification since unpublished data from *RIAgnosis* found no differences in oxytocin concentrations between extracted and simply evaporated saliva samples. RIA has been previously standardised and validated and represents the gold-standard for oxytocin measurement in biological fluids (*Kagerbauer et al., 2013*; *Landgraf, 1981*; *Landgraf and Günther, 1983*; *Landgraf et al., 1982a*; *Landgraf et al., 1983*; *Landgraf et al., 1982b*; *Martin et al., 2014*). The detection limit is in the 0.1–0.5 pg/sample range, depending on the age of the tracer. Cross-reactivity with vasopressin, ring moieties and terminal tripeptides of both oxytocin and vasopressin and a wide variety of peptides comprising 3 (alpha-melanocyte-stimulating hormone) up to 41 (corticotrophin-releasing factor) amino acids are <0.7% throughout. The intra- and inter-assay variabilities are <10% (*Kagerbauer et al., 2013*).

## Dataset A

### Between-visits reliability analysis

Reliability refers to the reproducibility of values of a measurement in repeated trials on the same individuals (*Hopkins, 2000*). Reliability can be quantified using two sets of metrics providing complementary information: absolute and relative reliability. Absolute reliability is the degree to which repeated measurements within the same subject vary over time (*Hopkins, 2000*). Relative reliability is the degree to which individuals maintain their position in a sample of subjects measured over time (*Hopkins, 2000*). Absolute and relative reliability of plasma and salivary oxytocin measurements were estimated using the within-subject CV and the ICC, respectively. ICC was estimated in a two-way mixed model, single measures, absolute agreement (*Koo and Li, 2016*). We first estimated the reliability across all four sessions, and subsequently for each pair of visits to assess if time-interval between sample collections may impact on reliability indexes estimation. Only participants presenting baseline measures across all four sessions were included in the reliability analysis, as previously suggested (*Cuesta Izquierdo and Fonseca Pedrero, 2014*).

Since there was considerable variability in the time-interval between visits across participants, we conducted a sensitivity analysis where we repeated our reliability analysis focusing on 15 pairs of consecutive measures that were collected with an exact time interval of 7 days between visits in 15 participants. Here, we recalculated the ICC and CV on this subset of our initial sample, using the approach described above. We also investigated whether within-participant variance in the time interval between sample acquisitions could predict within-participant variance in oxytocin concentrations across participants, using Spearman correlations.

Correlations are often used as an index for reliability, even though they cannot provide information about the absolute agreement of two sets of measurements (*Qin et al., 2019*). Hence, to facilitate comparisons with previous reports, we also calculated Pearson's correlation coefficients, with bootstrapping (1000 samples), to evaluate correlations between baseline concentrations of oxytocin for each pair of visits. The results of this analysis are presented below in the Table S1 and Figure S2.

### Within-visit reliability analysis (placebo visit)

To investigate the reliability of salivary and plasmatic oxytocin concentration within the same visit, we calculated the ICC and CV as described above for two samples acquired before any treatment administration and the intravenous infusion of saline during the placebo session. These samples where acquired with an approximate 15 min interval in between them.

### Mean concentrations across visits

Mean concentrations of saliva and plasma oxytocin across the four visits were compared using repeated-measures one-way analysis of variance.

### Treatment effects

The effect of treatment on blood/saliva oxytocin concentration were assessed using a $4 \times 2$ repeated-measures two-way analysis of variance Treatment (four levels: Spray, Nebuliser, Intravenous and Placebo) $\times$ Time (two levels: Baseline and post-administration). Post-hoc comparisons to clarify a significant interaction were corrected for multiple comparisons following the Tukey procedure.

### Association between salivary and plasmatic oxytocin levels

We assessed correlations between salivary and plasmatic concentrations of oxytocin sampled at baseline and post-administration. For the baseline measurements, we pooled data across treatment levels because there were no differences between groups on mean baseline concentrations of oxytocin. To account for the non-independence among the four data points within each subject, we used multilevel correlation, where we modeled participant as a random effect. For the post-administration measurements, we calculated Pearson's correlation coefficient for each treatment level separately because group differences in mean scores on these measures might result in illusory correlations if the drug and placebo samples were pooled together (*Paloyelis et al., 2010*). As a final sanity check, we also investigated correlations between the changes from baseline to post-administration in saliva and plasma in each of our treatment conditions separately. Since our sample was relatively small and therefore the lack of significant correlations between salivary and plasmatic oxytocin could simply reflect lack of sensitivity, we followed this frequentist correlation analysis with Bayesian statistics to quantify relative evidence for both the null and alternative hypotheses.

### Outliers and missing values

Salivary oxytocin concentrations were missing for three participants, and plasmatic oxytocin concentration for one participant. One measure of baseline oxytocin in saliva and two post-administration measures in the nebuliser condition were discarded after they had been identified as outliers. Outliers were identified using the *outlier labelling rule* (*Kwak and Kim, 2017*); this means that a data point was identified as an outlier if it was more than 1.5 x interquartile range above the third quartile or below the first quartile. A total of 13 and 16 participants were included in the reliability analysis of salivary and plasmatic oxytocin, respectively.

## Dataset B

### Mean concentrations across visits

Mean concentrations of plasma oxytocin across the two visits were compared using a paired T-test.

### Reliability analysis

Absolute and relative reliability of plasma oxytocin measurements were analysed for the two baseline measures obtained from each of the two visits, following the methods described for dataset A.

### Outliers and missing values

There were no missing values. One baseline measure for one of the visits was discarded after being identified as an outlier. A total of 19 participants were included in the reliability analysis.

Increasing the number of observations per individual and averaging across several samples collected on different occasions is an approach commonly used to control within-individual variation and maximize reliability (*Walker, 2008*). Hence, we expanded our reliability analysis by asking how many additional measures of the same individual would be theoretically required to achieve different levels of reliability of a hypothetical averaged measure, based on the initial reliabilities estimated in the studies A and B. This number was calculated using the Spearman-Brown prediction formula (*de Vet et al., 2017*). For these calculations, we considered cut-offs of ICC = 0.50 (fair reliability), 0.70 (moderate), and 0.80 (good) as suggested by *Koo and Li, 2016*.

## Statistical analysis

All statistical analyses were conducted on log-transformed oxytocin concentrations given the deviations of these measurements from a Gaussian distribution. The statistical analysis investigating

treatment effects on salivary and plasmatic oxytocin were performed using SPSS (version 24, IBM, Armonk, NY, USA). The frequentist and Bayesian correlations were implemented in the *correlation* package from R (version 3.5.3), using bootstrapping 1000 samples. For the bayesian correlations, we used beta priors' distributions centred around zero, with a width parameter of 1. An increase in Bayes Factor (BF) in our analyses corresponds to an increase in evidence in favor of the null hypothesis. To interpret BF, we used the Lee and Wagenmakers' classification scheme (*Lee MD, 2014*): BF <1/10, strong evidence for alternative hypothesis; 1/10 < BF < 1/3, moderate evidence for alternative hypothesis; 1/3 < BF < 1, anecdotal evidence for alternative hypothesis; BF >1, anecdotal evidence for the null hypothesis; 3 < BF < 10, moderate evidence for the null hypothesis; BF >10, strong evidence for the null hypothesis. Figures were produced using the *ggplot* package from R (version 3.5.3). p<0.05 (two-tailed) was set as threshold of statistical significance for all analyses.

## Acknowledgements

We thank Sofia Vasilakopoulou and Jack Loveridge for their assistance in data collection. We also thank Rosa Oliveira and Silia Vitoratou for their advice on statistical analysis. Most importantly, we thank all participants volunteering to both studies. This study was part-funded by: an Economic and Social Research Council Grant (ES/K009400/1) to YP; scanning time support by the National Institute for Health Research (NIHR) Biomedical Research Centre at South London and Maudsley NHS Foundation Trust and King's College London to YP; an unrestricted research grant by PARI GmbH to YP. Data collection for dataset B was supported by an IoPPN-MRC Excellence Studentship awarded to AG. MAM is in part supported by the National Institute for Health Research (NIHR) Biomedical Research Centre at South London and Maudsley NHS Foundation Trust and King's College London. Disclosures YP, DM, MM, and AG declare no competing financial interests. MM received research funding from Takeda and Lundbeck and support in kind from Johnson and Johnson and AstraZeneca. This manuscript represents independent research. The views expressed are those of the authors and not necessarily those of the NHS, the NIHR, the Department of Health and Social Care, or PARI GmbH.

## Additional information

### Funding

| Funder | Grant reference number | Author |
|---|---|---|
| Economic and Social Research Council | ES/K009400/1 | Yannis Paloyelis |
| NIHR Biomedical Research Centre, South London and Maudsley | | Mitul Mehta<br>Yannis Paloyelis |
| PARI GmbH | | Yannis Paloyelis |
| IoPPN, King's College London | | Anthony S Gabay |

The funders had no role in study design, data collection and interpretation, or the decision to submit the work for publication.

### Author contributions

Daniel Martins, Conceptualization, Data curation, Formal analysis, Investigation, Methodology, Writing - original draft, Writing - review and editing; Anthony S Gabay, Project administration, Writing - review and editing; Mitul Mehta, Supervision, Funding acquisition, Project administration, Writing - review and editing; Yannis Paloyelis, Conceptualization, Supervision, Funding acquisition, Project administration, Writing - review and editing

## Author ORCIDs

Daniel Martins  https://orcid.org/0000-0002-0239-8206
Anthony S Gabay  http://orcid.org/0000-0002-6946-1046
Yannis Paloyelis  https://orcid.org/0000-0002-4029-3720

## Ethics

Human subjects: All participants gave informed consent prior to testing. King's College London Research Ethics Committee approved the protocols for both studies (Dataset A: PNM/13/14-163; Dataset B: PNM/14/15-32).

## Decision letter and Author response

Decision letter https://doi.org/10.7554/eLife.62456.sa1
Author response https://doi.org/10.7554/eLife.62456.sa2

## Additional files

### Supplementary files

• Supplementary file 1. Table 1 – Absolute and relative between-visits reliability of oxytocin measurements in plasma and saliva for each pair of the four visits included in study A. ICC – Intraclass correlation coefficient; CV – coefficient of variation; CI – confidence interval; SD – Standard Deviation; *H0: ICC is not significantly different from 0; statistical significance was set to p<0.05 (two-tailed). N represents the actual size of the sample used to calculate the ICCs and the CVs. Table 2 – Absolute and relative between-visits reliability of oxytocin measurements in plasma and saliva controlling for time-interval between visits (dataset A). ICC – Intraclass correlation coefficient; CV – coefficient of variation; CI – confidence interval; SD – Standard Deviation; *H0: ICC is not significantly different from 0; statistical significance was set to p<0.05 (two-tailed). N represents the actual size of the sample used to calculate the ICCs and the CVs. Table 3 – Absolute and relative within-visit reliability of oxytocin measurements in plasma and saliva in the placebo visit of dataset A. ICC – Intraclass correlation coefficient; CV – coefficient of variation; CI – confidence interval; SD – Standard Deviation; *H0: ICC is not significantly different from 0; statistical significance was set to p<0.05 (two-tailed). N represents the actual size of the sample used to calculate the ICCs and the CVs.

• Source data 1.

• Transparent reporting form

## Data availability

All data generated or analysed during this study are included in the manuscript and supporting files.

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
