## [Decision Letter]

**Acceptance summary:**

Reviewers concurred that a single peripheral measurements of oxytocin at baseline may not provide valid trait markers of the physiology of the oxytocin system. The increases in salivary oxytocin observed after intranasal oxytocin most likely reflect unabsorbed peptide and should not be used by the field to predict treatment effects. *eLife* and reviewers appreciated the careful attention to methodology, which could influence interpretation or reinterpretation of many studies that rely on salivary and plasmatic oxytocin measurements.

**Decision letter after peer review:**

[Editors’ note: the authors submitted for reconsideration following the decision after peer review. What follows is the decision letter after the first round of review.]

Thank you for submitting your work entitled "Salivary and plasmatic oxytocin are not reliable biomarkers of the physiology of the oxytocin system in Humans" for consideration by *eLife*. Your article has been reviewed by three peer reviewers, and the evaluation has been overseen by a Reviewing Editor and a Senior Editor. The reviewers have opted to remain anonymous. Our decision has been reached after consultation between the reviewers. Based on these discussions and the individual reviews below, we regret to inform you that your work will not be considered further for publication in *eLife*.

The strengths of the study are the findings that a single oxytocin level measured from saliva or plasma is not meaningful in the way that the field might currently be measuring. The reviewers appreciated this finding, and the careful attention to detail, but felt that the results fell short of the level of insight required to meet the threshold for publication in *eLife*. In the setting where *eLife* decides not to proceed to publication, the reviews are returned in their unedited format for your benefit.

Reviewer #1:

This article describes the investigation of a valuable research question, given the interest in using salivary oxytocin measures as a proxy of oxytocin system activity. A strength of the study is the use of two independent datasets and the comparison between intranasal and intravenous administration. The authors report poor reliability for measuring salivary oxytocin across visits, that intravenous delivery does not increase concentrations, and that salivary and blood plasma concentrations are not correlated.

Introduction: While it's true that saliva collection provides logistical advantages, there are also measurement advantages (e.g., relatively clean matrix) that are summarised in the MacLean et al., 2019 study, which has already been cited.

It is important to note that the 1IU intravenous dose in this study led to equivalent concentrations in blood compared to intranasal administration

Materials and methods: When using both ELISA and HPLC-MS, extracted and unextracted samples are correlated when measuring oxytocin concentrations in saliva, at least in dogs (https://doi.org/10.1016/j.jneumeth.2017.08.033)

Statistical reporting: I ran the article through statcheck R package (a web version is also available) and found a number of inconsistencies with the reported statistics and their p values. For example, the authors reported: t(123) = 1.54, p = 0.41, but this should yield a p value of 0.13. The authors should do the same and fix these errors

Results: The confidence intervals for these correlations should be reported

Discussion, “Our observation of poor reliability for single measurements of plasma and saliva oxytocin raises questions about the interpretation of previous evidence seeking to associate single measurements of baseline oxytocin with individual differences in a range of neuro-behavioural or clinical traits…”: This is an important point, but it's important to note that the vast majority of these studies use plasma or saliva measures. Perhaps CSF measures are more reliable, but the question wasn't assessed in the present study, and I'm not sure if anyone has looked at this question.

Discussion final paragraph: I broadly agree with this conclusion, but it should be added that "single measurements of baseline levels of endogenous oxytocin in saliva and plasma are not stable under typical laboratory conditions" Perhaps these measures can be more stable using other means (i.e., better standardising collection conditions). But the fact remains, under typical conditions these measures do not demonstrate reliability

Reviewer #2:

To test questions whether salivary and plasmatic oxytocin at baseline reflect the physiology of the oxytocin system, and whether salivary oxytocin index its plasma levels, the authors quantified baseline plasmatic and/or salivary oxytocin using radioimmunoassay from two independent datasets. Dataset A comprised 17 healthy men sampled on four occasions approximately at weekly intervals. In the dataset A, oxytocin was administered intravenously and intranasally in a triple dummy, within-subject, placebo-controlled design and compared baseline levels and the effects of routes of administration. With dataset A, whether salivary oxytocin can predict plasmatic oxytocin at baseline and after intranasal and intravenous administrations of oxytocin were also tested. Dataset B comprised baseline plasma oxytocin levels collected from 20 healthy men sampled on two separate occasions. In both datasets, single measurements of plasmatic and salivary oxytocin showed insufficient reliability across visits (Intra-class correlation coefficient: 0.23-0.80; mean CV: 31-63%). Salivary oxytocin was increased after intranasal administration of oxytocin (40 IU), but intravenous administration (10 IU) does not significantly changes. Saliva and plasma oxytocin did not correlate at baseline or after administration of exogenous oxytocin (p>0.18). The authors suggest that the use of single measurements of baseline oxytocin concentrations in saliva and plasma as valid biomarkers of the physiology of the oxytocin system is questionable in men. Furthermore, they suggest that saliva oxytocin is a weak surrogate for plasma oxytocin and that the increases in saliva oxytocin observed after intranasal oxytocin most likely reflect unabsorbed peptide and should not be used to predict treatment effects.

The current study tested research questions relevant for the study field. The analyses in two independent datasets with different routes of oxytocin administrations is the strength of current study. However, the limited novelty of findings and several limitations are noticed in the current report as described below.

1) Previous study with similar results has already revealed that saliva oxytocin is a weak surrogate for plasmatic oxytocin, and increases in salivary oxytocin after the intranasal administration of exogenous oxytocin most likely represent drip-down transport from the nasal to the oral cavity and not systemic absorption (Quintana et al., 2018). Therefore, the novelty of current findings is limited. The authors should more clearly state the novelty of current results and the replication of previous findings.

2) As authors discussed in the limitation section of Discussion, the current study has several limitations such as analyses only in male participants and non-optimized timing of collection of saliva and blood due to the other experiments. These limitations are understandable, because the current study was the second analyses on the data of the other studies with the different aims. However, these limitations significantly limit the interpretations of the findings.

3) As reported the Materials and methods, the dataset A comprises administrations approximately 40 IU of intranasal oxytocin and 10 IU on intravenous. The rationale to set these doses should be described. Since the 40IU is different from 24 IU which is employed in most of the previous publications in the research filed, potential influence associated with the doses should be tested and discussed.

4) It is difficult to understand that no significant elevations in plasma oxytocin levels were observed after intranasal spray or nebuliser of oxytocin. From Figure 4A, the differences between levels at baseline and post administration are similar between nebuliser, spray, and placebo. Please discuss the potential interpretation on this result.

5) The reason why not to employ any correction for multipole comparisons in the statistical analyses should be clarified.

Reviewer #3:

Baseline samples of salivary and plasma oxytocin were assessed in 13, respectively, 16 participants, to assess intra-individual reliability across four time points (separated by approximately 8 days). The main results indicate that, while as a group, average salivary and plasma samples were not significantly different across time points, within-subject coefficient of variation (CV) and intra-class correlation coefficient (ICC) showed poor absolute and relative reliability of plasma and salivary oxytocin measurements over time. Also no association was established between plasma and salivary levels, either at baseline or after administration of oxytocin (either intranasally, or intravenously). Further, salivary/ plasma oxytocin was only enhanced after intranasal, respectively intravenous administration.

While the overall multi-session design seems solid, sample collections were performed in the context of larger projects and therefore there appear to be several limitations that reduce the robustness of the presented results and consequently the formulated conclusions.

General comments

1) A main conclusion of the current work is that “single measures of baseline oxytocin concentrations in saliva and plasma are not stable within the same individual”. It seems however that the study did not adhere to a sufficiently rigorous approach to put forward this conclusion. It lacks a control for several important factors, such as timing of the day at which saliva/ plasma samples were obtained, as well as sample volume.

Particularly while it is indicated that all visits were identical in structure, important information is missing with regard to whether or not sampling took place consistently at a particular point of time each day, to minimize the influence of circadian rhythm. Without this information it is not possible to draw any firm conclusions on the nature of the intra-individual variability as demonstrated in the salivary and plasma sampling.

Correspondingly, a deeper discussion is needed on the reason why ICC's were considerably variable across pairs of assessment sessions, with some pairs yielding good reliability, whereas others yielded (very) poor reliability. More detailed descriptions regarding sampling procedures (timing and sampling intervals) are necessary. Also, more information is needed on the volume of saliva collected at each session, to control for possible dilution effects.

2) It is indicated that the initial sample would allow to detect intra-class correlation coefficients (ICC) of at least 0.70 (moderate reliability) with 80% of power. Is this still the case after the drop-outs/ outlier removals? Since the main conclusions of the work rely on negative results (conclusions drawn from failures to reject the null hypothesis) it is important to establish the risk for false negatives within a design that is possibly underpowered.

3) Did the authors also assess within-session reliability? For example, by assessing ICC between pre and post-measurements in the placebo session.

4) It is indicated that the intra-assay variability of the adopted radioimmunoassay constitutes <10%. Were analyses of the current study run on duplicate samples? Was intra-assay variability assessed directly within the current sample?

Introduction and Discussion

5) The Introduction and Discussion is missing a thorough overview of previous studies assessing intra-individual variability in oxytocin levels.

6) The paper misses a discussion of previous studies addressing links between salivary/ plasma levels and central oxytocin (e.g. in cerebrospinal fluid). I understand the claim that salivary oxytocin cannot be used to form an estimate of systemic absorption, although technically, a lack of a link between salivary and plasma levels, does not necessarily imply a lack of a relationship to e.g. central levels. The lack of effect is limited to this specific relationship.

Materials and methods

7) Related to the general comment, the variability in days between sessions is relatively high (average 8.80 days apart (SD 5.72; range 3-28). However, it appears that no explicit measures were taken to control the conducted analyses for this variability.

8) A rationale for the adopted dosing and timing (115 min post administration) of the sample extraction is missing. Additionally, it seems that intravenous administrations were always given second, whereas intranasal administrations were given third, with a small delay of approximately 5 min. Hence, it seems that the timing of 115 min post-administration is only accurate for the intranasal administration.

9) Since the ICC of baseline samples showed poor reliability, it seems suboptimal to pool across sessions for assessing the relationship between salivary and blood measurements. It should be possible to perform e.g. partial correlations on the actual scores, thereby correcting for the repeated measure (subject ID). Further, since the sample size is relatively small (13 subjects), it might be recommended to use non-parametric (e.g. Spearmann correlations) instead of Pearson. The additional reporting of the Bayes factor is appreciated; it is very informative.

10) Now, the authors only compared relationships between salivary and plasma levels, either at baseline or post administration. I'm wondering whether it would be interesting to explore relationships between pre-to-post change scores in salivary versus plasma measures.

11) Please provide more information on the outlier detection procedure (outlier labelling rule).

12) Please indicate how deviations from a Gaussian distribution were assessed.

Results

13) Please verify the degrees of freedom for the post-hoc tests performed to assess pre-post changes at each treatment level (e.g. baseline vs Post administration: Spray – t(122) = 7.06, p < 0.001). Why is this 122? Shouldn't this be a simple paired-sample t-test with 13 subjects?

Reviewer #3:

• I would omit references to “biomarkers” or “biomarkers of the physiology of the oxytocin system” (e.g. in the title). I'm not sure which previous studies have claimed this link.

• In the last paragraph of the Introduction, only reference is made to assessing the effect of intravenous administration, whereas the study addresses both intravenous and intranasal effects.

• In the Materials and methods it is stated that “participants had no history of psychiatric disorders or substance abuse”. How was this assessed? Did the authors adopt any particular questionnaire or scale?

• Please indicate explicitly the number of subjects involved in any of the (Pearson) correlation analyses e.g. assessing relationships between salivary and plasma measurements.

• In the Discussion, it is indicated that “the time interval between measurements do not seem to significantly impact on the reliability of baseline oxytocin”. This was not explicitly assessed.

• Abbreviations in figures need to be reported in full in the figure legend.

• Figure 2. It would be helpful to use the same range for the y-axis of the plasma assessments (in sample A and B).

• It would be recommended to use a consistent order for reporting the different treatment levels. Now the table in Figure 1 uses the order spray, IV, placebo, Nebulizer, whereas in Figure 2, this order is changed to IV, nebulizer, placebo, spray. I guess the most logic order would be spray, nebulizer, IV and placebo (as used in-text).

• It is unclear why the presentation modes for presenting the associations between salivary and plasma OT at baseline (Figure 4) and post-administration (Figure 5) are different. It would be informative to plot for both the regression lines and distribution plots.

• Supplementary figures need figure legends.

• I would recommend omitting the reporting of “dataset B” from the main manuscript, or only report it (briefly) as a secondary analysis, with some additional information in the supplements.

[Editors’ note: further revisions were suggested prior to acceptance, as described below.]

Thank you for submitting the appeal for your article "Salivary and plasmatic oxytocin are not reliable biomarkers of the physiology of the oxytocin system in Humans" for consideration by *eLife*. The appeal has been reviewed by two of the original peer reviewers, and the evaluation has been overseen by Reviewing Editor Joseph Gleeson and Christian Büchel as the Senior Editor. The reviewers have opted to remain anonymous.

The reviewers are now more convinced by the data, and offer specific requests that, if completed, should allow the paper to proceed towards publication. The reviewers have discussed the reviews with one another and the Reviewing Editor has drafted this decision to help you prepare a revised submission.

Reviewer #1:

This is a revision of an article that I previously reviewed investigating if single peripheral measurements of baseline oxytocin in saliva and plasma are reliable trait markers of the physiology of the oxytocin system in humans.

The paper has now improved and most of my original queries have now been satisfactorily addressed.

However, I still have one comment regarding the author's response to query #2 "It is important to note that the 1IU intravenous dose in this study led to equivalent concentrations in blood compared to intranasal administration": I now better understand the justification for using a 10IU dose (i.e., "we demonstrate that even when plasmatic levels of OT are maintained substantially increased throughout the observation interval, we cannot detect increases in salivary oxytocin". However, this should also be better emphasized in the manuscript.

Reviewer #3:

Overall, the authors were able to provide additional information and conduct additional analyses that provided solutions to several of the raised methodological concerns (e.g. regarding timing of saliva collection, within-session reliability, sample size/ power, and other statistical remarks). Some other methodological issues may still need further clarification however, such as the possible impact of variability in collected sample volume across sessions (at least for the salivary samples). Also, considering the main research question of the current study (reliability of oxytocin sampling), it would be recommended that a measure of intra-assay variability was available for the own sample collections. The efforts for accounting for the effect of variability in between-session intervals are appreciated. I'm wondering however, whether it would be possible to perform secondary analyses on the whole data set (rather than performing subset analyses) regressing out possible effects of variability in between-session intervals.

In general, I feel that the manuscript already improved significantly compared to the initial submission, increasing its potential for publication tremendously. However, some of the raised concerns regarding the relative novelty regarding the study design and conclusions may still remain, raising questions whether a more specialized journal may be more suitable for its publication.

---

## [Author Response]

[Editors’ note: The authors appealed the original decision. What follows is the authors’ response to the first round of review.]

The strengths of the study are the findings that a single oxytocin level measured from saliva or plasma is not meaningful in the way that the field might currently be measuring. The reviewers appreciated this finding, and the careful attention to detail, but felt that the results fell short of the level of insight required to meet the threshold for publication in eLife. In the setting where eLife decides not to proceed to publication, the reviews are returned in their unedited format for your benefit.

We would like to thank you for the careful consideration of our manuscript and the three reviewers for their insightful comments. Overall, it seems to us the reviewers and editorial team found that the topic of our manuscript makes a valuable addition to the field. We have analysed in detail all reviewers’ comments and note that, overall, they are positive, fair and constructive. However, reading through the reviewers’ feedback, we felt that as a result of our insufficient clarity in communicating certain aspects of our work, the novelty and robustness of our findings might have been misunderstood. In particular, the reviewers raised concerns about the fact that we were not sufficiently strict in controlling major sources of variability that could have affected the levels of oxytocin in saliva and plasma between sessions, such as time of the day the samples were collected. This was not the case, since we used a consistent protocol where we controlled for time of day, food and liquid ingestion and pre-processing time of the samples, to minimize any potential influence from known sources of variability in oxytocin release to the periphery. The reviewers also raised concerns about reporting erroneous p-values and not having used correction for multiple comparisons where required, which as we explain in our responses below are not accurate and simply reflect lack of clarity in our report.

Prompted by the reviewers, we have now conducted some further analyses that strengthen the manuscript and add valuable insight into the interpretation of the data. For instance, we now show that the within-session reliability of salivary and plasmatic oxytocin in the placebo session is excellent. This strengthens our findings of low reliability across different days by suggesting that such poor reliability cannot simply be explained by measurement error or variation in the measurement conditions across sessions. We also conducted further analyses that discount variability in the time-interval between visits as a potential factor contributing to the low reliabilities we report in the manuscript.

Reviewer #1:This article describes the investigation of a valuable research question, given the interest in using salivary oxytocin measures as a proxy of oxytocin system activity. A strength of the study is the use of two independent datasets and the comparison between intranasal and intravenous administration. The authors report poor reliability for measuring salivary oxytocin across visits, that intravenous delivery does not increase concentrations, and that salivary and blood plasma concentrations are not correlated.Introduction: While it's true that saliva collection provides logistical advantages, there are also measurement advantages (e.g., relatively clean matrix) that are summarised in the MacLean et al., 2019, study, which has already been cited.

Thanks for the suggestion. We added this advantage:

“Compared to blood sampling, saliva collection presents several logistical and measurement advantages (i.e. relatively clean matrix)(1).”

It is important to note that the 1IU intravenous dose in this study led to equivalent concentrations in blood compared to intranasal administration

The reviewer is right that 10 IU (over 10min) in our case increased the concentrations of plasmatic oxytocin beyond those observed for the spray or nebuliser (we reported the full time-course of variations in plasmatic oxytocin in another manuscript we published earlier this year)*(2)*. This was an intentional aspect of our study design. We decided to use the highest intravenous dose (at the highest rate of 1IU/min) that we could get permission to administer safely in healthy volunteers as a proof of concept, so as to achieve a robust and prolonged increase in plasmatic oxytocin over the course of our full testing session. In this manner, we demonstrate that even when plasmatic levels of OT are maintained substantially increased throughout the observation interval, we cannot detect increases in salivary oxytocin. In this aspect, we believe that our manuscript goes one step beyond the important findings described in of Quintana et al. 2018(3), showing that this phenomenon is not linked to dosage (or to amount of increase in plasmatic levels of exogenous OT), as far as we can determine given the current safety standards for the administration of OT IV.

Please see also response to reviewer 2, point 1.

Materials and methods: When using both ELISA and HPLC-MS, extracted and unextracted samples are correlated when measuring oxytocin concentrations in saliva, at least in dogs (https://doi.org/10.1016/j.jneumeth.2017.08.033)

Thanks for pointing out this study. Indeed, in this specific study the authors found correlations between extracted and unextracted saliva samples. Such associations in humans have nevertheless been rare. In humans, the body of evidence suggests that the measurements obtained when comparing extracted samples to unextracted samples, or when comparing samples obtained using different methods of quantification (for instance, ELISA versus radioimmunoassay), do not correlate or show very low correlations (4, 5). Furthermore, most ELISA kits and HPLC-MS protocols to measure oxytocin have so far fallen short on sensitivity to detect the typical concentrations observed in humans at baseline (0-10pg/ml)(6). The current gold-standard method for quantifying oxytocin in biological fluids is the radioimmunoassay we used in this study(4). This method has shown superior sensitivity and specificity when compared to other quantification methods, when combined with extracted samples; therefore, it was our primary choice. We now highlight this advantage in the revised version of the manuscript more explicitly.

“For all analyses, we followed current gold-standard practices in the field and assayed oxytocin concentrations using radioimmunoassay in extracted samples, which has shown superior sensitivity and specificity when compared to other quantification methods(7).”

Statistical reporting: I ran the article through statcheck R package (a web version is also available) and found a number of inconsistencies with the reported statistics and their p values. For example, the authors reported: t(123) = 1.54, p = 0.41, but this should yield a p value of 0.13. The authors should do the same and fix these errors

Thanks very much for taking the time to check our statistical reporting thoroughly. We apologize if we were not sufficiently clear in the previous version of the manuscript, but the p-values we reported are corrected for multiple comparisons using Tukey correction. Currently, *statcheck* can only evaluate inconsistencies when the results are reported in the standard APA style and does not take into consideration corrections for multiple comparisons of any kind. We did check all of our statistical reporting and the p-values and correspondent statistics are correct (we only corrected an inadvertent error in reporting the degrees of freedom for these tests). In any case, we have now clarified in the manuscript when the reported p-values have been adjusted for multiple comparison to avoid any further confusion.

Results: The confidence intervals for these correlations should be reported

We have now added the confidence intervals, estimated using bootstrapping, in our Results section.

Discussion, “Our observation of poor reliability for single measurements of plasma and saliva oxytocin raises questions about the interpretation of previous evidence seeking to associate single measurements of baseline oxytocin with individual differences in a range of neuro-behavioural or clinical traits…”: This is an important point, but it's important to note that the vast majority of these studies use plasma or saliva measures. Perhaps CSF measures are more reliable, but the question wasn't assessed in the present study, and I'm not sure if anyone has looked at this question.

We are not aware of any study evaluating the stability of measurements of oxytocin in the CSF. Indeed, there are only a few studies sampling CSF to measure oxytocin in clinical patients and it is unlikely that CSF will become a widely used fluid to measure oxytocin in humans, given the invasiveness of the procedure to obtain CSF samples. Here, we wanted to refer specifically to saliva and plasma, which remain as the most popular options for measuring oxytocin in humans and which we investigated specifically in the current study. We have changed the text accordingly for clarity.

“Our data poses questions about the interpretation of previous evidence seeking to associate single measurements of baseline oxytocin in saliva and plasma with individual differences in a range of neuro-behavioural or clinical traits.”

Discussion final paragraph: I broadly agree with this conclusion, but it should be added that "single measurements of baseline levels of endogenous oxytocin in saliva and plasma are not stable under typical laboratory conditions" Perhaps these measures can be more stable using other means (i.e., better standardising collection conditions). But the fact remains, under typical conditions these measures do not demonstrate reliability

Thanks for the suggestion. We have revised the text accordingly throughout the manuscript (examples below). Our study is a pharmacological study, which means that it is conducted in a highly controlled setting and adheres to strict protocols (i.e. we tested participants at the same time of the day, we instructed participants to abstain from alcohol and heavy exercise for 24 h and from any beverage or food for 2 h before scanning). These exclusion criteria were stricter than those applied in a large number of studies sampling saliva and plasma for measuring oxytocin for the purposes estimating possible associations with various traits associating. Most of these studies do not control, for instance, for fluid or food ingestion. Therefore, we expected our reliability calculations to represent an optimistic estimate of the reliabilities of the salivary and plasmatic oxytocin concentration used in most studies.

For now, it remains unclear to us what factors might be driving the within-subject variability in salivary and plasmatic concentrations we report in this study. Thanks to reviewer 3, we are now confident that this is unlikely to represent measurement error (see response to reviewer 3, point 3).

“Here, we aimed to characterize the reliability of both salivary and plasmatic single measures of basal oxytocin in two independent datasets, to gain insight about their stability in typical laboratory conditions and their validity as trait markers for the physiology of the oxytocin system in humans.”

“In summary, single measurements of baseline levels of endogenous oxytocin in saliva and plasma as obtained in typical laboratory conditions are not stable and therefore their validity as trait markers of the physiology of the oxytocin system is questionable.”

Reviewer #2:To test questions whether salivary and plasmatic oxytocin at baseline reflect the physiology of the oxytocin system, and whether salivary oxytocin index its plasma levels, the authors quantified baseline plasmatic and/or salivary oxytocin using radioimmunoassay from two independent datasets. Dataset A comprised 17 healthy men sampled on four occasions approximately at weekly intervals. In the dataset A, oxytocin was administered intravenously and intranasally in a triple dummy, within-subject, placebo-controlled design and compared baseline levels and the effects of routes of administration. With dataset A, whether salivary oxytocin can predict plasmatic oxytocin at baseline and after intranasal and intravenous administrations of oxytocin were also tested. Dataset B comprised baseline plasma oxytocin levels collected from 20 healthy men sampled on two separate occasions. In both datasets, single measurements of plasmatic and salivary oxytocin showed insufficient reliability across visits (Intra-class correlation coefficient: 0.23-0.80; mean CV: 31-63%). Salivary oxytocin was increased after intranasal administration of oxytocin (40 IU), but intravenous administration (10 IU) does not significantly changes. Saliva and plasma oxytocin did not correlate at baseline or after administration of exogenous oxytocin (p>0.18). The authors suggest that the use of single measurements of baseline oxytocin concentrations in saliva and plasma as valid biomarkers of the physiology of the oxytocin system is questionable in men. Furthermore, they suggest that saliva oxytocin is a weak surrogate for plasma oxytocin and that the increases in saliva oxytocin observed after intranasal oxytocin most likely reflect unabsorbed peptide and should not be used to predict treatment effects.The current study tested research questions relevant for the study field. The analyses in two independent datasets with different routes of oxytocin administrations is the strength of current study. However, the limited novelty of findings and several limitations are noticed in the current report as described below.1) Previous study with similar results has already revealed that saliva oxytocin is a weak surrogate for plasmatic oxytocin, and increases in salivary oxytocin after the intranasal administration of exogenous oxytocin most likely represent drip-down transport from the nasal to the oral cavity and not systemic absorption (Quintana et al., 2018). Therefore, the novelty of current findings is limited. The authors should more clearly state the novelty of current results and the replication of previous findings.

We apologize for not describing the novelty and impact of our findings with sufficient clarity, and thanks for the opportunity to do so. Our study had two major goals. The first was to investigate whether single measurements of salivary and plasmatic concentrations of oxytocin can be reliably estimated within the same individual when collected at baseline conditions (i.e. without any experimental manipulation). As the reviewer highlighted, this is an important methodological question given the wide use of these measurements in a large and increasing number of studies to establish associations between the physiology of the oxytocin system and a number of brain and behavioural phenotypes in both clinical and non-clinical samples. However, to our knowledge, no previous study has appropriately conducted a thorough investigation of the reliability of these measurements (see also response to reviewer 3, point 5). Thanks to our study, we now know that when single measurements are collected at baseline, salivary and plasmatic oxytocin cannot provide a sufficiently stable trait marker of the physiology of the oxytocin system in humans. As we highlight in the manuscript, this finding should deter the field from making strong claims based exclusively on associations of phenotypes with single measurements of peripheral oxytocin concentrations. Furthermore, our study also describes two very concrete implications of our findings which we believe are very important for the field. First, if baseline level of OT is to be used as a trait marker, future studies should, as much as possible, rely on repeated measures within the same participant but collected on different days to maximize reliability. Second, this less than perfect reliability should be taken into consideration when calculating the sizes of the samples needed to detect a certain effect, if it exists, with sufficient statistical power.

The second goal of our study was, as pointed out by the reviewer, to revisit the findings of Quintana et al., 2018(3), but this time with two major design modifications which could strengthen the conclusions from that study.

The first modification was the dose of intravenous oxytocin administered, which was considerably higher (see response to reviewer 1, point 2). The administration of a higher dose that resulted in substantial and sustained increases in plasmatic oxytocin throughout the two hours observation period can only strengthen the previous conclusion that increases in plasmatic oxytocin cannot be detected in salivary measurements, and that this is not a matter of dose (as far as we can ascertain by administering the maximum intravenous dose we could safely administer in healthy volunteers). We believe that this is an important addition to the literature.

The second modification regarded the choice of the method we used to quantify oxytocin. In this study, we used radioimmunoassay, which is superior to ELISA in sensitivity and hence more appropriate to measure the low concentrations of oxytocin in saliva and plasma typically detected in humans at baseline conditions (1-10 pg/ml; for most individuals 1-5 pg/ml)(6). For instance, in Quintana et al., 2018(3) the limitations in the sensitivity of the ELISA kit used led the authors to discard around 50% of the collected saliva samples. Hence, our study replicates and extends the previous findings from Quintana et al., 2018 in important ways, demonstrating that the lack of an association between increases plasmatic oxytocin and salivary measurements is not limited by the dose of intravenous oxytocin administered or limitations of the sensitivity of the method used to quantify oxytocin.

We have now made the novelty and contribution of our work more explicit:

“Currently, we lack robust evidence that single measures of endogenous oxytocin in saliva and plasma at rest are stable enough to provide a valid trait marker of the activity of the oxytocin system in healthy individuals. […] Such evidence is urgently required, given reports that plasma and saliva levels of oxytocin are frequently altered during neuropsychiatric illness and that they co-vary with clinical aspects of disease(13).”

“Our findings were not consistent with these expectations. We could replicate previous evidence that intravenous oxytocin does not increase salivary oxytocin(3) and extended it by showing that the lack of increase in salivary oxytocin is not limited to the specific low dose of intravenous OT that was previously used (1IU) and that it is not driven by the insufficient sensitivity of the OT measurement method (which had resulted in more than 50% of the saliva samples being discarded in the previous study(3).”

2) As authors discussed in the limitation section of Discussion, the current study has several limitations such as analyses only in male participants and non-optimized timing of collection of saliva and blood due to the other experiments. These limitations are understandable, because the current study was the second analyses on the data of the other studies with the different aims. However, these limitations significantly limit the interpretations of the findings.

Here, we would like to highlight two aspects. First, most studies in the field are indeed conducted in men to avoid potential confounding from fluctuations in oxytocin concentrations across the menstrual cycle in women. Therefore, our study is representative of the typical samples used in most human studies. Second, we did not optimize our study to collect repeated samples of saliva. Indeed, it would have been interesting to describe the full-time course of variations of oxytocin concentrations in saliva after intranasal and intravenous administration. However, this does not detract the importance of our findings in respect to our first aim (which was our main goal).

We agree with the reviewer though that it is at least theoretically possible that we could have missed the window for increases in salivary oxytocin after intravenous oxytocin if it existed, given that we only sampled one post-administration time-point. However, we believe this was unlikely for one reason. Despite the sustained increase (throughout the two-hour observation interval) in plasmatic oxytocin following the intravenous administration of oxytocin, we observed no increase in salivary oxytocin post-dosing (at ~115 min). Unless the half-life of oxytocin is shorter in saliva than in the blood (which we do not know yet), we expected the levels of salivary oxytocin to mirror the changes in plasma – potentially with a slight delay given the time that it might take for oxytocin concentrations to build up in saliva through ultrafiltration from the blood, but this was not the case. Most likely the half-life of oxytocin in the saliva is not shorter than in the blood, since a previous study found increased concentrations of oxytocin in saliva up to 7h after administration of intranasal oxytocin (as the reviewer pointed out below, in our study we no longer could detect significant increases in plasmatic oxytocin after the intranasal administration of 40 IU with two different methods at around 115 mins post-administration). Therefore, while we acknowledge these limitations we also believe they do not detract from the importance of our main findings and the potential they hold to influence the field towards a more rigorous use of these measurements. Please see below for the implemented changes in the text.

“It is possible that we may have missed peak increases in saliva oxytocin after the intravenous administration of exogenous oxytocin if they occurred between treatment administration and post-administration sampling. This is unlikely given that the dose we administered intravenously resulted in sustained increases in plasmatic oxytocin over the course of two hours. Unless the half-life of oxytocin in saliva is much shorter than in the plasma, it would be surprising to not find any increases in salivary oxytocin after intravenous oxytocin given that concentrations of oxytocin in the plasma were still elevated at the specific time-point of our second saliva sample. Currently, we have no estimate for the half-life of oxytocin in saliva; however, given that previous studies have found evidence of increased salivary oxytocin after single intranasal administrations of 16IU and 24IU oxytocin up to seven hours post-administration(19), it is unlikely that the half-life of oxytocin is shorter in the saliva than in the plasma.”

3) As reported the Materials and methods, the dataset A comprises administrations approximately 40 IU of intranasal oxytocin and 10 IU on intravenous. The rationale to set these doses should be described. Since the 40IU is different from 24 IU which is employed in most of the previous publications in the research filed, potential influence associated with the doses should be tested and discussed.

Thank you for the opportunity to clarify this aspect of our work. With respect of our primary aims (to investigate whether single measurements of salivary and plasmatic oxytocin at baseline can be reliably measured within individuals across different days), the choice of doses is of course not relevant.

With respect to our secondary aim, namely, to investigate whether salivary oxytocin can be used to index concentrations of oxytocin in the plasma, particularly after the administration of synthetic oxytocin using the intranasal and intravenous routes, the administered doses are relevant.

The data reported here were collected as part of a larger project – which determined the choice of both intranasal and IV doses *(2)*. As explained in our response to reviewer 1, point 2, the selection 10IU (over 10min) was the highest intravenous dose that we could get permission to administer safely in healthy volunteers as a proof of concept, so as to achieve a robust and prolonged increase in plasmatic oxytocin over the course of our full testing session. In this manner, we demonstrate that even when plasmatic levels of OT are maintained substantially increased throughout the observation interval, we cannot detect increases in salivary oxytocin.

Regarding the intranasal OT dose, it is worth noting that the 24 IU is indeed popular in oxytocin studies, but not exclusive, and generally the selection of dose in oxytocin studies has not been informed by detailed dose-response characterizations. Our choice of 40IU was made for the purposes of matching our previous work on the pharmacodynamics of OT in healthy volunteers(20), and is a dose we (21-29) and others (e.g. (30)) have commonly used with patients.

A potentially important implication if dose variations also imply variation in the total volume of liquid administered (as is usually the case with standard nasal sprays – but not with the nebuliser), then it is likely that the potential for drip-down might increase for higher volumes and decrease for lower volumes. As far as we know, no study has ever investigated the impact of administered volume on salivary oxytocin after the intranasal administration of synthetic oxytocin, but we agree this would be an important point to look at. We have now expanded our Discussion to accommodate this point.

“We expect this phenomenon to be particularly pronounced for higher administered volumes. Further studies should examine the impact of different administered volumes on increases in salivary oxytocin.”

4) It is difficult to understand that no significant elevations in plasma oxytocin levels were observed after intranasal spray or nebuliser of oxytocin. From Figure 4A, the differences between levels at baseline and post administration are similar between nebuliser, spray, and placebo. Please discuss the potential interpretation on this result.

The plasmatic concentrations of oxytocin we report in this study refer solely to the samples acquired at around 2h after the administration of intranasal oxytocin. We reported the full-time course of changes in plasmatic oxytocin in a paper published earlier this year*(2) –* which we now refer the reader to. We did find increases in plasmatic oxytocin after administration of oxytocin with the spray and nebuliser (around 3x the baseline concentrations) that did not differ between intranasal methods of administration. Plasmatic oxytocin reached a peak within 15 mins from the end of the intranasal administrations. Given the short half-life of oxytocin in the plasma, we believe it is not surprising that at 115 mins after the end of our last treatment administration the concentrations of oxytocin in the plasma are no longer different from the placebo condition.

“The full time course of changes in plasmatic oxytocin after the administration of intranasal and intravenous oxytocin in this study has been reported elsewhere(2).”

5) The reason why not to employ any correction for multipole comparisons in the statistical analyses should be clarified.

We apologize that this was not sufficiently clear, but we did correct for multiple testing using the Tukey procedure in our analyses investigating the effects of treatment on salivary and plasmatic oxytocin (this was described in Treatment effects). If the reviewer meant something else, we would be glad to follow any further advice on multiple testing correction he/she might have.

“Treatment effects: The effect of treatment on blood/saliva oxytocin concentration were assessed using a 4 x 2 repeated-measures two-way analysis of variance Treatment (four levels: Spray, Nebuliser, Intravenous and Placebo) x Time (two levels: Baseline and post-administration). Post-hoc comparisons to clarify a significant interaction were corrected for multiple comparisons following the Tukey procedure.”

Reviewer #3:Baseline samples of salivary and plasma oxytocin were assessed in 13, respectively, 16 participants, to assess intra-individual reliability across four time points (separated by approximately 8 days). The main results indicate that, while as a group, average salivary and plasma samples were not significantly different across time points, within-subject coefficient of variation (CV) and intra-class correlation coefficient (ICC) showed poor absolute and relative reliability of plasma and salivary oxytocin measurements over time. Also no association was established between plasma and salivary levels, either at baseline or after administration of oxytocin (either intranasally, or intravenously). Further, salivary/ plasma oxytocin was only enhanced after intranasal, respectively intravenous administration.While the overall multi-session design seems solid, sample collections were performed in the context of larger projects and therefore there appear to be several limitations that reduce the robustness of the presented results and consequently the formulated conclusions.General comments1) A main conclusion of the current work is that “single measures of baseline oxytocin concentrations in saliva and plasma are not stable within the same individual”. It seems however that the study did not adhere to a sufficiently rigorous approach to put forward this conclusion. It lacks a control for several important factors, such as timing of the day at which saliva/ plasma samples were obtained, as well as sample volume.Particularly while it is indicated that all visits were identical in structure, important information is missing with regard to whether or not sampling took place consistently at a particular point of time each day, to minimize the influence of circadian rhythm. Without this information it is not possible to draw any firm conclusions on the nature of the intra-individual variability as demonstrated in the salivary and plasma sampling.

Thanks for pointing this out. Indeed, we were not sufficiently explicit on how strict we were in controlling for some potential sources of variability that could have contributed to the lack of reliability we report here. Our data was acquired in the context of two human pharmacological studies, which by design were strict on a number of aspects to minimize unwarranted noise. All participants were tested in the same period of the day (morning) to avoid the potential contribution of circadian fluctuations of oxytocin. In dataset A, we tried, as much as possible, to match the exact time participants were tested between visits, using the start time of the first visit as a reference. With the exception of one participant, where one session was conduct 1h and 30 mins later than the other three, all the remaining participants from study A were tested within 1h of the exact start time of session 1. Further, we also instructed participants to abstain from alcohol and heavy exercise for 24 h and from any beverage or food for 2 h before scanning. Hence, we believe our sampling protocol was strict enough to discard any potential contribution of major known sources of variability in oxytocin levels.

The reviewer also inquiries about the volume of the samples. For the plasma samples, we used a standardized protocol and collected the same blood volume in all participants, visits and time-points (1 EDTA tube of approximately 4 ml). The saliva samples were collected using *Salivettes.* Participants were instructed to place the swab from the *Salivette* kit in their mouth and chew it gently for 1 min to soak as much saliva as possible. After this, the swab was then returned back to the *Salivette* and centrifuged. In both cases, to avoid degradation of the peptide in the collected sample, we followed a strict protocol where all samples were put immediately in iced water until centrifugation, which happened within 20 mins of sample collection. Samples were then immediately stored at -80°C until analysis. Hence, differences in degradation of the peptide related to the processing of the sample are also unlikely to justify the poor reliabilities we report here.

For completeness, we have now added all of these further details to our Materials and methods section.

“All visits were conducted during the morning to avoid the potential confounding of circadian variations in oxytocin levels(31, 32). In addition, we also made sure that each participant was tested at approximately the same time across all four visits (all participants were tested in sessions with less than one hour difference in their onset time, except for one participant where the difference in the onset of one session compared to the other three sessions was 1.5h). “

“Blood was collected in ethylenediaminetetraacetic acid vacutainers (Kabe EDTA tubes 078001), placed in iced water and centrifuged at 1300 × g for 10 minutes at 4°C within 20 minutes of collection and then immediately pipetted into Eppendorf vials. Samples were immediately stored -80°C until analysis. […] We followed this strict protocol, putting all samples in iced water until centrifugation with immediate storage at -80°C until analysis to minimize the impact putative differences in degradation of the peptide related to differences in the processing of the samples might have on the reliability of the estimated concentrations of oxytocin.”

Correspondingly, a deeper discussion is needed on the reason why ICC's were considerably variable across pairs of assessment sessions, with some pairs yielding good reliability, whereas others yielded (very) poor reliability.

Currently we have no insightful hypothesis on why this could have been the case. Indeed, we found higher ICCs for only 2 out of 6 pairs of visits for the plasma. However, it is plausible that this might have occurred by chance. In any case, we should note that the 95% confidence intervals for the ICCs of our different pairs of samples overlap; this suggests that there is no evidence that the ICCs we estimated for the specific two pairs where we found higher reliabilities are significantly higher than those observed in the remaining pairs.

“If there are specific reasons explaining the higher reliability indices observed for the specific pairs of sessions, these reasons remain to be elucidated. However, it is not implausible that we might have found higher reliabilities for these specific two pairs by chance, since the 95% confidence intervals for the ICCs for all pairs of samples overlapped.”

More detailed descriptions regarding sampling procedures (timing and sampling intervals) are necessary. Also, more information is needed on the volume of saliva collected at each session, to control for possible dilution effects.

This information has been added to the revised version of the manuscript (please see response to your point number 1). As a further clarification, oxytocin concentrations were measured in plasma and saliva aliquots of 0.5 ml, following the standard operating procedures of *RIAgnosis*. This volume was used for all participants, sessions and time-points. Furthermore, for measuring cortisol, the *salivettes* were shown to allow for an almost 100% recovery, regardless of cortisol concentration, volume of the sample or method of quantification(33), suggesting that the sampling method is robust.

2) It is indicated that the initial sample would allow to detect intra-class correlation coefficients (ICC) of at least 0.70 (moderate reliability) with 80% of power. Is this still the case after the drop-outs/ outlier removals? Since the main conclusions of the work rely on negative results (conclusions drawn from failures to reject the null hypothesis) it is important to establish the risk for false negatives within a design that is possibly underpowered.

We understand the concern of the reviewer. However, according to the power calculations provided by Bujang and Baharum, 2017(34), the four repeated samples we collected in Dataset A would have allowed us to detect an ICC of 0.5 with 80% of statistical power even with only 13 subjects (which is the lowest sample size we used for the analysis on saliva in dataset A). The two samples we collected in Dataset B would allow us to detect an ICC of 0.6 with 80% of statistical power even with only 19 subjects. Hence, both datasets were powered to detect an ICC of 0.7 with acceptable power, if it existed, even after the exclusion of outliers.

3) Did the authors also assess within-session reliability? For example, by assessing ICC between pre and post-measurements in the placebo session.

Thanks for the suggestion. Indeed, we had not performed this analysis before but we agree it would be informative. We calculated the ICC and CV for the two samples acquired before any treatment administration and the intravenous infusion of saline during the placebo session. These samples where acquired with an approximate 15 min interval in between them. In this analysis, we found that the ICC was excellent 0.92 and the CV 20%. This additional analysis strengthens our findings by supporting the idea that our poor reliabilities across different days reflect true biological variability and cannot be attributed to measurement error. These new findings have now been included in the revised version of the manuscript.

Abstract

“Results: Single measurements of plasmatic and salivary oxytocin showed poor reliability across visits in both datasets. The reliability was excellent when samples were collected within 15 minutes from each other in the placebo visit.”

“Within-visit reliability analysis: To investigate the reliability of salivary and plasmatic oxytocin concentration within the same visit, we calculated the ICC and CV as described above for two samples acquired before any treatment administration and the intravenous infusion of saline during the placebo session. These samples where acquired with an approximate 15 minutes interval in between them.”

“Furthermore, in a further analysis assessing the within-session stability of plasmatic oxytocin using two measurements collected 15 minutes apart from each other in the placebo visit (one sample collected at baseline and the other after the intravenous administration of saline), we found excellent within-session reliability (ICC=0.92, CV=20%). Together, this suggests that the low reliability of endogenous oxytocin measurements across visits in the current study results from true intrinsic individual biological variability and not technical variability/error in the method used for oxytocin quantification.”

4) It is indicated that the intra-assay variability of the adopted radioimmunoassay constitutes <10%. Were analyses of the current study run on duplicate samples? Was intra-assay variability assessed directly within the current sample?

We reported the intra-assay variability determined by *RIAgnosis* during the development of this assay(35). This was not specifically assessed for the current study.

Introduction and Discussion5) The Introduction and Discussion is missing a thorough overview of previous studies assessing intra-individual variability in oxytocin levels.

Thanks for the suggestion. We have now included in our Introduction/Discussion an overview of previous studies attempting to tackle this question, which unfortunately do not address this question with sufficient detail or using the appropriate methods and statistical analyses (see response to reviewer 2, point 1). Hence, from the available evidence, it is not possible to draw robust conclusions about the validity of concentrations of oxytocin in saliva and plasma as valid trait markers of the activity of the oxytocin system. With this manuscript, we hope we can prompt further discussion and guide the field towards a more rigorous use of these measurements. A thorough discussion of this literature has now been added to the Introduction and Discussion.

“Our observation of poor reliability questions the use of single measurements of baseline oxytocin concentrations in saliva and plasma as valid trait markers of the physiology of the oxytocin system in humans. […] Hence, our estimates of reliability are a better starter point for all studies where specific circumstances potentially affecting the activity of the oxytocin system have not been specified a priori.”

6) The paper misses a discussion of previous studies addressing links between salivary/ plasma levels and central oxytocin (e.g. in cerebrospinal fluid). I understand the claim that salivary oxytocin cannot be used to form an estimate of systemic absorption, although technically, a lack of a link between salivary and plasma levels, does not necessarily imply a lack of a relationship to e.g. central levels. The lack of effect is limited to this specific relationship.

In this study, we did not intend to investigate whether salivary and plasmatic oxytocin are valid proxies for the activity of the oxytocin system in the brain. Our data does not address that question and a thorough discussion of these studies falls, in our opinion, out of the scope of the manuscript. Instead, we focused on whether measurements of oxytocin in saliva and plasma (by far the most commonly used biological fluids to measure oxytocin) are sufficiently stable to provide valid indicators of the physiology of the oxytocin system in humans. Additionally, we also investigated whether salivary oxytocin can index plasmatic oxytocin at baseline and after the administration of synthetic oxytocin using different routes of administration.

A previous meta-analysis of studies correlating peripheral and CSF measurements of oxytocin has shown that most likely peripheral and CSF measurements do not correlate at baseline; significant correlations could be found after intranasal administration of oxytocin or specific experimental manipulations, such as stress(37). We believe that currently we still do not have a clear answer about the extent to which these peripheral fluids can actually index oxytocin concentrations in the brain (even if associations with CSF are evident in specific instances). For instance, no study has ever shown that CSF oxytocin actually predicts the concentrations of oxytocin in the extracellular fluid of the brain. Given what we currently know about the synaptic release of oxytocin in the brain(38) (in contrast with former theories of exclusive bulk diffusion in the CSF(39)), we think we have good reasons to suspect this might not be the case.

The only contribution our study can make in that respect is highlighting our current lack of understanding of how oxytocin reaches saliva if not from the blood. Currently there is no evidence of direct secretion of oxytocin to the saliva (not from acinar secretion or nerve terminals release). Hence, as it stands, the most likely mechanism for oxytocin to entry the saliva is from the blood (for instance, by ultrafiltration). If increases in plasmatic oxytocin after intravenous oxytocin cannot produce any significant increases in salivary oxytocin (shown in ours and in a previous study), how does oxytocin reach the saliva and why might it be able to predict concentrations in the CSF, if it does? In this respect, we hope our study highlights the need for further research shedding light on the mechanisms underlying these potential saliva – CSF relationships, if they exist. We would be glad to accommodate any other hypothesis the reviewer might have on this respect.

“The lack of increase in salivary oxytocin after the intravenous administration of exogenous oxytocin that was consistently found in our study and in a previous study(3) also raises the question of how oxytocin reaches the saliva if not from the blood. Currently there is no evidence of direct acinar secretion or direct nerve terminals release of oxytocin to the saliva; therefore, transport from the blood remains as the most plausible mechanism of appearance of oxytocin in the saliva. Clarifying these mechanisms of transport is paramount, given the current hypothesis that salivary oxytocin might be superior to plasma in indexing central levels of oxytocin in the CSF(40).”

Materials and methods7) Related to the general comment, the variability in days between sessions is relatively high (average 8.80 days apart (SD 5.72; range 3-28). However, it appears that no explicit measures were taken to control the conducted analyses for this variability.

Thanks for point this out. Indeed, we were not sufficiently thorough in exploring the impact of this potential variability in the time gap between visits on our estimated ICCs. Thanks to the reviewer we now acknowledged this limitation of our analysis and decided to explore this further. We decided to run the following sensitivity analysis. First, we went back to our dataset A and identified all pairs of consecutive measures that were collected with an exact time interval of 7 days between visits. We could retrieve 15 examples of these pairs from 15 different participants for both saliva and plasma. Then, we recalculated the ICC and CV on this subset of our initial sample. In line with our main analysis, we found poor reliabilities for both salivary and plasmatic oxytocin; in both cases the ICCs were not significantly different from 0 and the CVs were 49% and 40%, respectively. This further analysis has been added to the revised version of the manuscript. We hope the reviewer shares our vision that our main conclusion of poor reliabilities of single measurements of baseline oxytocin in saliva and plasma cannot be simply attributed to the variability in the number of days between visits.

“Since there was considerable variability in the time-interval between visits across participants, we conducted a sensitivity analysis where we repeated our reliability analysis focusing on 15 pairs of consecutive measures that were collected with an exact time interval of 7 days between visits in 15 participants. Here, we recalculated the ICC and CV on this subset of our initial sample, using the approach described above.”

“These poor reliabilities are unlikely to be explained by variability in the time-interval between visits of the same individual, since we also found poor reliability indexes for both saliva and plasma when we restricted our analysis to a subset of our sample controlling for the exact number of days spacing visits.”

8) A rationale for the adopted dosing and timing (115 min post administration) of the sample extraction is missing. Additionally, it seems that intravenous administrations were always given second, whereas intranasal administrations were given third, with a small delay of approximately 5 min. Hence, it seems that the timing of 115 min post-administration is only accurate for the intranasal administration.

We collected saliva samples before any treatment administration and after the end of our scanning session (collection of saliva samples in between was just not possible because the participants were inside the MRI machine and could not have moved their heads). For the plasma, we collected samples before any treatment administration, after each treatment administration and at other five time-points during the scanning session. Here, we only report the plasma data that was acquired concomitantly with the saliva samples (the full-time course of plasma changes in plasmatic oxytocin has been reported elsewhere*(2)*).

In the manuscript, we report post-administration times from the end of the full treatment administration protocol. Hence, as the reviewer highlights our post-administration sample was collected at around 115 mins from the last intranasal administration and 120 mins from the end of the intravenous administration. We have now made this aspect explicit in the revised version of the manuscript.

“For the purposes of this report, we use the plasmatic and salivary oxytocin measurements that were obtained at baseline and at 115 minutes after the end of our last treatment administration (this means that our post-administration samples were collected 115 mins after the intranasal administrations and 120 mins after the intravenous administration of oxytocin).”

9) Since the ICC of baseline samples showed poor reliability, it seems suboptimal to pool across sessions for assessing the relationship between salivary and blood measurements. It should be possible to perform e.g. partial correlations on the actual scores, thereby correcting for the repeated measure (subject ID). Further, since the sample size is relatively small (13 subjects), it might be recommended to use non-parametric (e.g. Spearmann correlations) instead of Pearson. The additional reporting of the Bayes factor is appreciated; it is very informative.

Thanks for the suggestion. In fact, for the correlation the reviewer mentions we indeed used a multilevel approach where we specified subject as a random effect. This allowed us to deal with the dependence of measurements coming from the same subject in different visits. Furthermore, since we also had concerns about the sample size, we calculated Pearson correlations but used bootstrapping (1000 samples) to obtain the 95% confidence intervals and assess significance. Bootstrapping is a robust statistical technique which allows significance testing independently of any assumptions about the distribution of the data and is robust to outliers. Please see subsection “Association between salivary and plasmatic oxytocin levels”.

10) Now, the authors only compared relationships between salivary and plasma levels, either at baseline or post administration. I'm wondering whether it would be interesting to explore relationships between pre-to-post change scores in salivary versus plasma measures.

Thanks for the suggestion. We have now conducted this further analysis and we could not find any significant correlation between changes from baseline to post-administration in any of our treatment conditions. As for our other correlation analyses, here we also conducted Bayesian inference, which supported the idea that the null hypothesis of no significant correlation between changes in saliva and plasma from baseline to post-administration is at least 4x more likely than the alternative hypothesis. This further analysis strengthens our confidence that changes in salivary oxytocin after administration of oxytocin using the intranasal and intravenous routes should not be used to predict systemic absorption to the plasma.

“As a final sanity check, we also investigated correlations between the changes from baseline to post-administration in saliva and plasma in each of our treatment conditions separately.”

“Furthermore, we could not find any significant correlation between changes in salivary or plasmatic oxytocin from baseline to 115 mins after the end of our last treatment administration in any of our four treatment conditions. The lack of significant associations between salivary and plasmatic oxytocin (and respective changes from baseline) was further supported through our Bayesian analyses which demonstrated that given our data the null hypotheses were at least three times more likely than the alternative hypothesis.”

11) Please provide more information on the outlier detection procedure (outlier labelling rule).

This information has now been added to the revised version of the manuscript.

“Outliers were identified using the outlier labelling rule(41); this means that a data point was identified as an outlier if it was more than 1.5 x interquartile range above the third quartile or below the first quartile.”

12) Please indicate how deviations from a Gaussian distribution were assessed.

We used the combined assessment of i) differences between mean and median; ii) skewness and kurtosis; iii) histogram; iv) Q-Q plots; and v) the Kolmogorov-Smirnov and Shapiro-Wilk normality tests. Deviations from a normal distribution is common in the concentration of several analytes in the saliva (42), including oxytocin (15); hence, following the current recommendations, we used log transformations of the raw concentrations but plot the raw concentrations to facilitate the interpretation of our plots.

Results13) Please verify the degrees of freedom for the post-hoc tests performed to assess pre-post changes at each treatment level (e.g. baseline vs Post administration: Spray – t(122) = 7.06, p < 0.001). Why is this 122? Shouldn't this be a simple paired-sample t-test with 13 subjects?

We apologize for this oversight. Indeed, we did a mistake in copying the values of the degrees of freedom from SPSS. We have now corrected these values. All the other p-values and F or T values were reported correctly and hence are not changed in the revised version of the manuscript (please see also response to reviewer 1, question 4 regarding inconsistencies in the reported p-values).

Reviewer #3:• I would omit references to “biomarkers” or “biomarkers of the physiology of the oxytocin system” (e.g. in the title). I'm not sure which previous studies have claimed this link.

We have revised the text accordingly throughout the manuscript. Some examples below.

Title:

“Are single peripheral measurements of baseline oxytocin in saliva and plasma reliable trait markers of the physiology of the oxytocin system in humans?”

Abstract:

“However, questions remain about whether they are sufficiently stable to provide valid trait markers of the physiology of the oxytocin system, and whether salivary oxytocin can accurately index its plasmatic concentrations.”

“Conclusions: Our findings question the use of single measurements of baseline oxytocin concentrations in saliva and plasma as valid trait markers of the physiology of the oxytocin system in humans and suggest that, at best, these measurements can provide reliable state markers.”

Introduction

“Here, we investigate the second assumption, which is a prerequisite if single measurements of baseline levels of endogenous oxytocin are to be used as a valid trait markers of the physiology of the human oxytocin system(43).”

• In the last paragraph of the Introduction, only reference is made to assessing the effect of intravenous administration, whereas the study addresses both intravenous and intranasal effects.

Thanks for pointing that out. Indeed, we did not make full justice to the fact that our study investigated two different routes of administration (intranasal vs intravenous) and two intranasal methods (spray versus nebuliser). This information has been added to the Introduction in the revised version of the manuscript.

“Here, we aimed to characterize the reliability of both salivary and plasmatic single measures of basal oxytocin in two independent datasets, to gain insight about their stability in typical laboratory conditions and their validity as trait markers for the physiology of the oxytocin system in humans. Additionally, we investigated whether salivary oxytocin concentration reflects plasmatic oxytocin by examining i) if the intravenous administration of exogenous oxytocin increases the concentration of salivary oxytocin; ii) how potential changes in salivary oxytocin compare between different routes of administration (intranasal versus intravenous) and methods of intranasal administration (spray versus a nebuliser); and iii) the correlation between plasmatic and salivary oxytocin levels at baseline and after the administration of exogenous oxytocin using two different methods of intranasal administration (spray versus nebuliser) and the intravenous route.”

• In the Materials and methods it is stated that “participants had no history of psychiatric disorders or substance abuse”. How was this assessed? Did the authors adopt any particular questionnaire or scale?

We apologized for not having described this in sufficient detail. In dataset A, we screened participants for psychiatric conditions using the Symptom Checklist-90-Revised(44) and the Beck Depression Inventory-II(45) questionnaires. In dataset B, participants were screened using the MINI International Neuropsychiatric Interview(46). This information has been added to the revised version of the manuscript.

“All participants had no history of psychiatric disorders or substance abuse, scored negatively on a screening test for recreational drug use, and did not currently use any medication. In dataset A, we screened participants for psychiatric conditions using the Symptom Checklist-90-Revised(44) and the Beck Depression Inventory-II(45) questionnaires. In dataset B, we used the MINI International Neuropsychiatric Interview(46).”

• Please indicate explicitly the number of subjects involved in any of the (Pearson) correlation analyses e.g. assessing relationships between salivary and plasma measurements.

The exact number of subjects used for each correlation analysis has now been added to the revised version of the manuscript.

• In the Discussion, it is indicated that “the time interval between measurements do not seem to significantly impact on the reliability of baseline oxytocin”. This was not explicitly assessed.

Indeed, we did not explain in detail how we evaluated this aspect. This conclusion derives from the comparison of the overlap of the 95% confidence intervals for the ICCs estimated for each pair of sessions. As the reviewer can see in Supplementary file 2, there is considerable overlap between the 95% CI of the different ICCs. This suggests that there are no significant differences between ICCs across different pairs of visits, even if we find numerically higher ICCs for the following plasma pairs visits 1-2 and visits 3-4. We now explain this in further detail in the revised version of our Discussion.

“The time-interval between measurements do not seem to significantly impact on the reliability of baseline oxytocin, as suggested by the overlap of the 95% confidence intervals of the ICCs estimated for each pair of sessions.”

• Abbreviations in figures need to be reported in full in the figure legend.

Thanks for pointing that out. We have revised the legend of Figure 1 to report the abbreviations missing.

• Figure 2. It would be helpful to use the same range for the y-axis of the plasma assessments (in sample A and B).

We have implemented the suggestion in the revised version of Figure 2.

• It would be recommended to use a consistent order for reporting the different treatment levels. Now the table in Figure 1 uses the order spray, IV, placebo, Nebulizer, whereas in Figure 2, this order is changed to IV, nebulizer, placebo, spray. I guess the most logic order would be spray, nebulizer, IV and placebo (as used in-text).

Thanks for pointing out this inconsistency. In the revised version of our figures and tables we now use a consistent order, which matches the order we used in-text.

• It is unclear why the presentation modes for presenting the associations between salivary and plasma OT at baseline (Figure 4) and post-administration (Figure 5) are different. It would be informative to plot for both the regression lines and distribution plots.

We decided to not include the marginal distribution plots in Figure 5 to make the figure easier to read. We have now revised Figure 5 to match the mode of presentation of Figure 4.

• Supplementary figures need figure legends.

These legends were presented at the end of the manuscript as recommended by *eLife*. *eLife* does not accept the inclusion of text in supplementary.

• I would recommend omitting the reporting of “dataset B” from the main manuscript, or only report it (briefly) as a secondary analysis, with some additional information in the supplements.

Thanks for the suggestion, but as explained in the point just above, *eLife* does not accept the inclusion of text in supplementary.

[Editors’ note: what follows is the authors’ response to the second round of review.]

Reviewer #1:This is a revision of an article that I previously reviewed investigating if single peripheral measurements of baseline oxytocin in saliva and plasma are reliable trait markers of the physiology of the oxytocin system in humans.The paper has now improved and most of my original queries have now been satisfactorily addressed.However, I still have one comment regarding the author's response to query #2 "It is important to note that the 1IU intravenous dose in this study led to equivalent concentrations in blood compared to intranasal administration": I now better understand the justification for using a 10IU dose (i.e., "we demonstrate that even when plasmatic levels of OT are maintained substantially increased throughout the observation interval, we cannot detect increases in salivary oxytocin". However, this should also be better emphasized in the manuscript.

Thank you for your previous helpful comments, we are glad we were able to address your comments and that you find that the manuscript has improved. Thank you also for the opportunity to clarify this remaining point. We have now highlighted this aspect of our design in the Introduction and Materials and methods sections of the revised version of our manuscript (see below).

“Additionally, we investigated whether salivary oxytocin concentration reflects plasmatic oxytocin by examining i) if the intravenous administration of a large dose of oxytocin which produces sustained increases in plasmatic oxytocin over the course of two hours also increases the concentration of salivary oxytocin;”<bold />

“The administration of 10IU of oxytocin intravenously produces sustained increases in the levels of plasmatic oxytocin over a two hours course(1). This aspect of our design allows us to eliminate the possibility that the lack of changes in salivary oxytocin is due to under-dosing.”

Reviewer #3:Overall, the authors were able to provide additional information and conduct additional analyses that provided solutions to several of the raised methodological concerns (e.g. regarding timing of saliva collection, within-session reliability, sample size/ power, and other statistical remarks). Some other methodological issues may still need further clarification however, such as the possible impact of variability in collected sample volume across sessions (at least for the salivary samples). Also, considering the main research question of the current study (reliability of oxytocin sampling), it would be recommended that a measure of intra-assay variability was available for the own sample collections. The efforts for accounting for the effect of variability in between-session intervals are appreciated. I'm wondering however, whether it would be possible to perform secondary analyses on the whole data set (rather than performing subset analyses) regressing out possible effects of variability in between-session intervals.In general, I feel that the manuscript already improved significantly compared to the initial submission, increasing its potential for publication tremendously. However, some of the raised concerns regarding the relative novelty regarding the study design and conclusions may still remain, raising questions whether a more specialized journal may be more suitable for its publication.

Thank you for your previous helpful suggestions and comments. We are glad that the additional information and analyses addressed most of the methodological concerns and that you consider that the manuscript has substantially improved. Thank you also for the opportunity to clarify any remaining concerns, which we address below.

As a further clarification of the novelty of our study, we would like to highlight that our study is the first of its kind to evaluate the question of the validity of single peripheral measurements of baseline oxytocin in saliva and plasma as reliable trait markers of the physiology of the oxytocin system in humans using the appropriate methods for the quantification of oxytocin concentrations and the appropriate statistical approach to assessing reliability. Thanks to our study, we came to appreciate that most likely previous claims that single measurements of baseline oxytocin in saliva and plasma are sufficiently stable within individuals to provide a valid trait marker of the physiology of the oxytocin system were probably misguided. Our second question related to the validity of salivary oxytocin to index oxytocin concentrations in the plasma is indeed an extension of previous work; here, critical aspects of our design were novel and allowed us to expand previous findings in important ways (namely, the use of a high dose of intravenous oxytocin which eliminated the possibility that the lack of an association might have been driven by under-dosing, and a method for oxytocin quantification that was sufficiently sensitive to the baseline physiological range of oxytocin).

Regarding the specific points raised:

1) Possible impact of variability in collected sample volume across sessions (at least for the salivary samples).

We have now provided further details in the manuscript that clarify the standard operating procedures for blood sample and saliva collection. In both cases we followed the standard operating procedures as provided by Professor Rainer Landgraf (RIAgnosis, standard operating procedures available upon request).

Blood samples: The collected volume of blood was the same across participants and sessions since we used standard 5ml EDTA vacutainer tubes. Following centrifuging, exactly 0.5 ml plasma was aliquoted in 2ml Eppendorf vials (and stored until analysis as described in the manuscript).

Saliva: We collected saliva using *Salivettes* (Sarstedt 51.1534.500), which were then centrifuged and exactly 0.5ml aliquots of saliva was pipetted to 1.5ml Eppendorf vials and stored until analysis. This was the same across participants and sessions. As expected when collecting saliva with *Salivettes*, small variations in the initial volume of saliva collected before centrifugation may have existed. However, we do not think this could induce considerable artificial variability in the concentrations of salivary oxytocin between sessions. First, we measured concentrations of oxytocin (pg of oxytocin per millilitre) in the exact same centrifuged volume for all participants and sessions. The *salivette* allows for a recovery of mean saliva volumes in the range of 1.1 ± 0.3 ml, which is at least 2 x higher than the minimal volume we would need to quantify oxytocin. Previous studies have only raised concerns about the recovery of the concentrations of small peptides in saliva samples collected with *salivettes* when the initial collected volumes are lower than 0.25 ml(2, 3), which was not the case in our study. We have now included these clarifications in the revised version of the manuscript (see below).

“Blood was collected in 5ml ethylenediaminetetraacetic acid vacutainers (Kabe EDTA tubes 078001), placed in iced water and centrifuged at 1300 × g for 10 minutes at 4°C within 20 minutes of collection and then 0.5ml of plasma was immediately pipetted into 2ml Eppendorf vials. […] Minimizing the time-interval samples were kept in the collection devices also allowed us to keep potential absorption to the walls of these recipients to a minimum(2).”

2) Obtaining a measure of intra-assay variability for own sample collections: We agree with the reviewer, and thanks to their previous suggestion we estimated variability in our own sample using two sample acquisitions that were obtained within 15 mins from each other in the placebo visit. Our within-session reliability analysis showed excellent reliability (ICC=0.92, CV=20%). We hope that the reviewer agrees that this is reassuring that the poor reliability across sessions we report here is unlikely to have resulted simply from measurement error.

3) The efforts for accounting for the effect of variability in between-session intervals are appreciated. I'm wondering however, whether it would be possible to perform secondary analyses on the whole data set (rather than performing subset analyses) regressing out possible effects of variability in between-session intervals.

Thank you for this suggestion and your appreciation of our effort to account for the effect of variability in between-session intervals. We understand that the reviewer wonders to what extent variability in the within-subject intervals between samples might contribute to the lack of reliability that we report. Following discussions with colleagues and a statistician, we addressed this questions through three possible lines of analyses, which all provided converging results:

a) The previously reported sensitivity analyses. The sensitivity analysis we presented in the last version of the manuscript, where we examined the reliabilities for both salivary and plasmatic oxytocin in a subset of our sample where two consecutive saliva and plasma samples were collected with an exact gap of seven days. For both plasma and saliva, the estimated ICCs were not significantly different from 0 and the CVs were 40% and 49%, respectively.

b) The previously reported pairwise comparisons (e.g. visit 1 vs visits 2/3/4), where the interval between sample acquisition may not have been exactly fixed, but it systematically increased with each visit. As we reported, we did not find any significant effect of time-interval on our estimated ICCs. If time-interval was driving the poor reliabilities, then we would have expected that in our pairwise analyses reliability would be consistently higher for samples closer in time and drop as the time-interval between sessions increases. This was not what we found (please see Between-visits reliability analysis for each pair of visits in our manuscript).

c) Additionally, we conducted a new analysis using information from the whole sample, as the reviewer suggested. Here, we explored the potential impact of variability in the time interval between sessions as follows. The ICC is the ratio of between-participant variance to total variance, which in turn is the sum of within- and between-participant variance. Therefore, one possible way to address the contribution of variability in the within-subject intervals between samples is to examine the relationship between within-participant variance in oxytocin concentrations and within-participant variance in the time interval between sample acquisitions, across participants. We did this and found that correlations between oxytocin and time interval variances were non-significant for both plasma (Spearman Rho = 0.406, *p* = 0.118) and salivary measurements (Spearman Rho = -0.524, *p* = 0.065). (For salivary measurements, it may be worth noting the negative sign of the correlations, suggesting that increased variability in time interval would predict, if significant, decreased variability in oxytocin measurements, which is counterintuitive. We know better of course than interpreting non-significant results, especially when inconsistent in direction which suggests random estimates around a mean of 0 association between variances).

We believe that these three lines of analysis provide converging evidence that variability in the time intervals between acquisitions is unlikely to be driving the poor ICCs we report in this manuscript (see below for the changes in the revised version of the Discussion). This converging evidence is consistent with our main conclusion that single peripheral measurements of oxytocin at baseline may not provide valid trait markers of the physiology of the oxytocin system and we that hope the reviewer agrees that the variability in time-interval (at least within the range of days examined in this study) between sessions is unlikely to be a key driver of the low reliabilities we report here.

**“**Variance in the within-subject intervals between samples did not correlate with within-participant variance in oxytocin concentrations across participants neither for plasma (Spearman Rho = 0.406, p = 0.118) or saliva (Spearman Rho = -0.524, p = 0.065).”

**“**These poor reliabilities are unlikely to be explained by variability in the time-interval between visits of the same individual. Three lines of converging evidence support this conclusion. First, we also found poor reliability indexes for both saliva and plasma when we restricted our analysis to a subset of our sample controlling for the exact number of days spacing visits. Second, we did not find any significant effect of time-interval on our estimated ICCs. If time-interval was driving the poor reliabilities, then we would have expected that in our pairwise analyses reliability would be consistently higher for samples closer in time and drop as the time-interval between sessions increases. This was not what we found. Third, variability in the within-subject intervals between samples did not correlate with within-participant variance in oxytocin concentrations across participants.”

**“**Since there was considerable variability in the time-interval between visits across participants, we conducted a sensitivity analysis where we repeated our reliability analysis focusing on 15 pairs of consecutive measures that were collected with an exact time interval of 7 days between visits in 15 participants. Here, we recalculated the ICC and CV on this subset of our initial sample, using the approach described above. We also investigated whether within-participant variance in the time interval between sample acquisitions could predict within-participant variance in oxytocin concentrations across participants, using Spearman correlations.”

References

1. Groschl M (2008): Current status of salivary hormone analysis. Clin Chem. 54:1759-1769.

2. Martins DA, Mazibuko N, Zelaya F, Vasilakopoulou S, Loveridge J, Oates A, et al. (2020): Effects of route of administration on oxytocin-induced changes in regional cerebral blood flow in humans. Nat Commun. 11:1160.

3. Quintana DS, Westlye LT, Smerud KT, Mahmoud RA, Andreassen OA, Djupesland PG (2018): Saliva oxytocin measures do not reflect peripheral plasma concentrations after intranasal oxytocin administration in men. Horm Behav. 102:85-92.

4. Szeto A, McCabe PM, Nation DA, Tabak BA, Rossetti MA, McCullough ME, et al. (2011): Evaluation of enzyme immunoassay and radioimmunoassay methods for the measurement of plasma oxytocin. Psychosom Med. 73:393-400.

5. Lefevre A, Mottolese R, Dirheimer M, Mottolese C, Duhamel JR, Sirigu A (2017): A comparison of methods to measure central and peripheral oxytocin concentrations in human and non-human primates. Sci Rep. 7:17222.

6. Jong TR, Menon R, Bludau A, Grund T, Biermeier V, Klampfl SM, et al. (2015): Salivary oxytocin concentrations in response to running, sexual self-stimulation, breastfeeding and the TSST: The Regensburg Oxytocin Challenge (ROC) study. Psychoneuroendocrinology. 62:381-388.

7. McCullough ME, Churchland PS, Mendez AJ (2013): Problems with measuring peripheral oxytocin: can the data on oxytocin and human behavior be trusted? Neurosci Biobehav Rev. 37:1485-1492.

8. Leng G, Ludwig M (2016): Intranasal Oxytocin: Myths and Delusions. Biol Psychiatry. 79:243-250.

9. Onodera M, Ishitobi Y, Tanaka Y, Aizawa S, Masuda K, Inoue A, et al. (2015): Genetic association of the oxytocin receptor genes with panic, major depressive disorder, and social anxiety disorder. Psychiatr Genet. 25:212.

10. Verhagen M, Verweij KJH, Lodder GMA, Goossens L, Verschueren K, Van Leeuwen K, et al. (2020): A SNP, Gene, and Polygenic Risk Score Approach of Oxytocin-Vasopressin Genes in Adolescents' Loneliness. J Res Adolesc. 30 Suppl 2:333-348.

11. Valstad M, Alvares GA, Egknud M, Matziorinis AM, Andreassen OA, Westlye LT, et al. (2017): The correlation between central and peripheral oxytocin concentrations: A systematic review and meta-analysis. Neurosci Biobehav Rev. 78:117-124.

12. Crockford C, Deschner T, Ziegler TE, Wittig RM (2014): Endogenous peripheral oxytocin measures can give insight into the dynamics of social relationships: a review. Front Behav Neurosci. 8:68.

13. Rutigliano G, Rocchetti M, Paloyelis Y, Gilleen J, Sardella A, Cappucciati M, et al. (2016): Peripheral oxytocin and vasopressin: Biomarkers of psychiatric disorders? A comprehensive systematic review and preliminary meta-analysis. Psychiatry Res. 241:207-220.

14. Feldman R, Gordon I, Influs M, Gutbir T, Ebstein RP (2013): Parental oxytocin and early caregiving jointly shape children's oxytocin response and social reciprocity. Neuropsychopharmacology. 38:1154-1162.

15. Schneiderman I, Zagoory-Sharon O, Leckman JF, Feldman R (2012): Oxytocin during the initial stages of romantic attachment: relations to couples' interactive reciprocity. Psychoneuroendocrinology. 37:1277-1285.

16. Gordon I, Pratt M, Bergunde K, Zagoory-Sharon O, Feldman R (2017): Testosterone, oxytocin, and the development of human parental care. Horm Behav. 93:184-192.

17. Zhu LG, Sun LP, Fan FY, Zhang DQ, Li CQ, Wang DQ (2019): Stability of plasma proteins and factors in Chinese universal pooled plasma. J Int Med Res. 47:2637-2646.

18. Gordon I, Zagoory-Sharon O, Leckman JF, Feldman R (2010): Oxytocin and the development of parenting in humans. Biol Psychiatry. 68:377-382.

19. van Ijzendoorn MH, Bhandari R, van der Veen R, Grewen KM, Bakermans-Kranenburg MJ (2012): Elevated Salivary Levels of Oxytocin Persist More than 7 h after Intranasal Administration. Front Neurosci. 6:174.

20. Paloyelis Y, Doyle OM, Zelaya FO, Maltezos S, Williams SC, Fotopoulou A, et al. (2016): A Spatiotemporal Profile of In Vivo Cerebral Blood Flow Changes Following Intranasal Oxytocin in Humans. Biol Psychiatry. 79:693-705.

21. Martins D, Davies C, De Micheli A, Oliver D, Krawczun-Rygmaczewska A, Fusar-Poli P, et al. (2020): Intranasal oxytocin increases heart-rate variability in men at clinical high risk for psychosis: a proof-of-concept study. Transl Psychiatry. 10:227.

22. Schmidt A, Davies C, Paloyelis Y, Meyer N, De Micheli A, Ramella-Cravaro V, et al. (2020): Acute oxytocin effects in inferring others' beliefs and social emotions in people at clinical high risk for psychosis. Transl Psychiatry. 10:203.

23. Martins D, Leslie M, Rodan S, Zelaya F, Treasure J, Paloyelis Y (2020): Investigating resting brain perfusion abnormalities and disease target-engagement by intranasal oxytocin in women with bulimia nervosa and binge-eating disorder and healthy controls. Transl Psychiatry. 10:180.

24. Leslie M, Leppanen J, Paloyelis Y, Treasure J (2020): A pilot study investigating the influence of oxytocin on attentional bias to food images in women with bulimia nervosa or binge eating disorder. J Neuroendocrinol. 32:e12843.

25. Leslie M, Leppanen J, Paloyelis Y, Nazar BP, Treasure J (2019): The influence of oxytocin on risk-taking in the balloon analogue risk task among women with bulimia nervosa and binge eating disorder. J Neuroendocrinol. 31:e12771.

26. Davies C, Rutigliano G, De Micheli A, Stone JM, Ramella-Cravaro V, Provenzani U, et al. (2019): Neurochemical effects of oxytocin in people at clinical high risk for psychosis. European neuropsychopharmacology : the journal of the European College of Neuropsychopharmacology. 29:601-615.

27. Davies C, Paloyelis Y, Rutigliano G, Cappucciati M, De Micheli A, Ramella-Cravaro V, et al. (2019): Oxytocin modulates hippocampal perfusion in people at clinical high risk for psychosis. Neuropsychopharmacology. 44:1300-1309.

28. Leslie M, Leppanen J, Paloyelis Y, Treasure J (2019): The influence of oxytocin on eating behaviours and stress in women with bulimia nervosa and binge eating disorder. Molecular and cellular endocrinology. 497:110354.

29. Paloyelis Y, Krahe C, Maltezos S, Williams SC, Howard MA, Fotopoulou A (2016): The Analgesic Effect of Oxytocin in Humans: A Double-Blind, Placebo-Controlled Cross-Over Study Using Laser-Evoked Potentials. J Neuroendocrinol. 28.

30. Cai Q, Feng L, Yap KZ (2018): Systematic review and meta-analysis of reported adverse events of long-term intranasal oxytocin treatment for autism spectrum disorder. Psychiatry Clin Neurosci. 72:140-151.

31. Amico JA, Levin SC, Cameron JL (1989): Circadian rhythm of oxytocin in the cerebrospinal fluid of rhesus and cynomolgus monkeys: effects of castration and adrenalectomy and presence of a caudal-rostral gradient. Neuroendocrinology. 50:624-632.

32. Reppert SM, Perlow MJ, Artman HG, Ungerleider LG, Fisher DA, Klein DC (1984): The circadian rhythm of oxytocin in primate cerebrospinal fluid: effects of destruction of the suprachiasmatic nuclei. Brain Res. 307:384-387.

33. Powell J, DiLeo T, Roberge R, Coca A, Kim JH (2015): Salivary and serum cortisol levels during recovery from intense exercise and prolonged, moderate exercise. Biol Sport. 32:91-95.

34. Bujang MA, Baharum N (2017): A simplified guide to determination of sample size requirements for estimating the value of intraclass correlation coefficient: a review. Arch Orofac Sci. 12:1-11.

35. Landgraf R (1981): Simultaneous measurement of arginine vasopressin and oxytocin in plasma and neurohypophyses by radioimmunoassay. Endokrinologie. 78:191-204.

36. Bui E, Hellberg SN, Hoeppner SS, Rosencrans P, Young A, Ross RA, et al. (2019): Circulating levels of oxytocin may be elevated in complicated grief: a pilot study. European Journal of Psychotraumatology. 10.

37. MacLean EL, Wilson SR, Martin WL, Davis JM, Nazarloo HP, Carter CS (2019): Challenges for measuring oxytocin: The blind men and the elephant? Psychoneuroendocrinology. 107:225-231.

38. Knobloch HS, Grinevich V (2014): Evolution of oxytocin pathways in the brain of vertebrates. Front Behav Neurosci. 8:31.

39. Leng G, Ludwig M (2008): Neurotransmitters and peptides: whispered secrets and public announcements. The Journal of physiology. 586:5625-5632.

40. Martin J, Kagerbauer SM, Gempt J, Podtschaske A, Hapfelmeier A, Schneider G (2018): Oxytocin levels in saliva correlate better than plasma levels with concentrations in the cerebrospinal fluid of patients in neurocritical care. J Neuroendocrinol.e12596.

41. Kwak SK, Kim JH (2017): Statistical data preparation: management of missing values and outliers. Korean J Anesthesiol. 70:407-411.

42. Miller R, Plessow F (2013): Transformation techniques for cross-sectional and longitudinal endocrine data: application to salivary cortisol concentrations. Psychoneuroendocrinology. 38:941-946.

43. Wang SJ, Cohen N, Katz DA, Ruano G, Shaw PM, Spear B (2006): Retrospective validation of genomic biomarkers-- what are the questions, challenges and strategies for developing useful relationships to clinical outcomes-- workshop summary. Pharmacogenomics J. 6:82-88.

44. Ruis C, van den Berg E, van Stralen HE, Huenges Wajer IM, Biessels GJ, Kappelle LJ, et al. (2014): Symptom Checklist 90-Revised in neurological outpatients. J Clin Exp Neuropsychol. 36:170-177.

45. Sacco R, Santangelo G, Stamenova S, Bisecco A, Bonavita S, Lavorgna L, et al. (2016): Psychometric properties and validity of Beck Depression Inventory II in multiple sclerosis. Eur J Neurol. 23:744-750.

46. Sheehan DV, Lecrubier Y, Sheehan KH, Amorim P, Janavs J, Weiller E, et al. (1998): The Mini-International Neuropsychiatric Interview (M.I.N.I.): the development and validation of a structured diagnostic psychiatric interview for DSM-IV and ICD-10. The Journal of clinical psychiatry. 59 Suppl 20:22-33;quiz 34-57.